# DS-LLM: Leveraging Dynamical Systems to Enhance Both Training and Inference of Large Language Models

**Ruibing Song[1], Chuan Liu[1], Chunshu Wu[1], Ang Li[2], Dongfang Liu[3], Yingnian Wu[4], Tony (Tong) Geng[1]**

Department of Electrical and Computer Engineering, University of Rochester[1]
Physical & Computational Sciences Directorate, Pacific Northwestern National Laboratory[2]
Department of Computer Engineering, Rochester Institute of Technology[3]
Department of Statistics and Data Science, University of California, Los Angeles[4]

## Abstract

The training of large language models (LLMs) faces significant computational cost challenges, limiting their scalability toward artificial general intelligence (AGI) and broader adoption. With model sizes doubling approximately every 3.4 months and training costs escalating from \$64 million for GPT-4 in 2020 to \$191 million for Gemini Ultra in 2023, the economic burden has become unsustainable. While techniques such as quantization offer incremental improvements, they fail to address the fundamental computational bottleneck. In this work, we introduce DS-LLM, a novel framework that leverages dynamical system (DS)-based machines, which exploit Natural Annealing to rapidly converge to minimal energy states, yielding substantial efficiency gains. Unlike traditional methods, DS-LLM maps LLM components to optimization problems solvable via Hamiltonian configurations and utilizes continuous electric current flow in DS-machines for hardware-native gradient descent during training. We mathematically demonstrate the equivalence between conventional LLMs and DS-LLMs and present a method for transforming a trained LLM into a DS-LLM. Experimental evaluations across multiple model sizes demonstrate orders-of-magnitude improvements in speed and energy efficiency for both training and inference while maintaining consistent accuracy. Additionally, we provide an in-depth analysis of the challenges and potential solutions associated with this emerging computing paradigm, aiming to lay a solid foundation for future research.

## 1 Introduction

Large language models (LLMs) are propelling rapid advancements in AI, with model sizes doubling approximately every 3.4 months. This exponential growth necessitates unprecedented computational resources for both training and inference, with training costs soaring from \$64 million for GPT-4 in 2020 to \$191 million for Gemini Ultra in 2023. As a result, a significant portion of cloud and HPC resources is now dedicated to LLMs, raising societal, environmental, and energy concerns. Meanwhile, Moore's Law is losing momentum, exacerbating the situation. While optimizing traditional computing methods remains essential, the demand for new, energy-efficient architectures is growing. The challenge is to sustainably scale AI models, particularly LLMs, without incurring prohibitive costs.

Several alternative computing paradigms have emerged, including quantum computing (Kerenidis et al., 2024), optical computing (Anderson et al., 2024), and computing-in-memory (Tu et al., 2023). While promising, these technologies face significant technical barriers and require further development. In the near term, is it feasible to rely on mature CMOS-based technology to accelerate LLM training from 10 million hours to 10,000 hours while reducing energy consumption from 20 terajoules to 200 megajoules?

Recent advances in dynamical system-based (DS) machines provide a compelling solution. DS-machines incorporate an electrodynamics-based model that naturally converges to the minimum energy state by following the physical dynamics of electrons. By correctly setting the initial state,

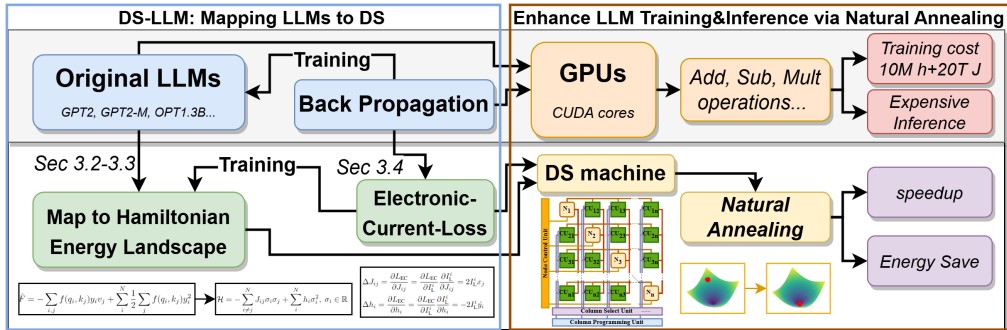

Figure 1: Overview of DS-LLM.

boundary conditions, and final state, various problems can be embedded into this "Natural Annealing" process. As no external energy is theoretically required during such a process, DS-machines are extremely energy efficient and have already demonstrated remarkable potential in some real-world applications. For example, solving complex optimization problems like MAX-CUT (Hamerly et al., 2019), working on graph learning problems including traffic predictions, air quality, taxi demand, and pandemic progression (Wu et al., 2024). Given their CMOS-compatible analog implementation, DS-machines operate at room temperature, achieving remarkable efficiency gains. Previous research has demonstrated over $1,000\times$ speedup and $100,000\times$ energy reduction for graph learning tasks while improving accuracy compared to state-of-the-art Graph Neural Networks (GNNs) running on commercial GPUs.

Can DS-machines be leveraged to accelerate LLMs? Unfortunately, no prior work has successfully mapped existing deep learning models like LLMs onto DS-machines. Previous approaches (Wu et al., 2024) have ignored the architectural structure of neural networks, instead directly fitting data to the energy landscape of DS-machines. However, for complex tasks such as natural language processing (NLP), directly modeling the energy landscape of a DS system would be prohibitively challenging. Instead, could we re-purpose well-researched LLM architectures to guide the energy landscape formation, effectively enabling DS-machines to execute LLM computations through Natural Annealing?

To address this challenge, we introduce Dynamical System-based Large Language Model (DS-LLM), the first algorithmic framework that bridges LLMs and DS-machines. Our approach constructs the energy landscape of DS-machines based on a reference LLM, ensuring they can accurately reproduce LLM outputs while leveraging Natural Annealing for inference. Additionally, we propose an Electric-Current-Loss-based (ECL) training method, which exploits the continuously evolving electric currents in DS-machines as hardware-native gradients to optimize model parameters dynamically. We mathematically prove the equivalence of DS-LLM and traditional LLMs for both inference (Sections 3.2-3.3) and training (Section 3.4), and we empirically validate our approach using models ranging from GPT-2-124M to LLaMA-2-7B. Our experimental results demonstrate orders-of-magnitude speedup and energy reduction while preserving accuracy.

This work represents an early-stage exploration into leveraging dynamical systems for efficient LLM computation. Our analysis is supported by theoretical proofs and empirical simulations conducted using standard commercial analog hardware simulation flows. While further research is needed to assess real-world performance using fabricated DS hardware, our results highlight significant potential. We hope this work inspires the community to further investigate DS-based architectures as a promising avenue for sustainable LLM scaling.

The main contributions of this work are summarized as follows:

- We propose DS-LLM, the first framework that unlocks the extraordinary inherent computational power of dynamical systems to revolutionize LLM computational efficiency.
- We introduce an online continuous training method that enables DS-Machine to perform instant on-device lifelong training, extending its remarkable computational efficiency from AI test to training.
- Experimental results across models from GPT-2-124M to LLaMA-2-7B on five downstream datasets show that, on average, DS-Machine achieves a $5.3\times10^3$ speedup for training and $2.4\times10^2$ for inference, along with a reduction in energy consumption of $2.3\times10^5$ during training and $6.4\times10^3$ during inference.

## 2 BACKGROUND AND PRELIMINARY KNOWLEDGE

### 2.1 DYNAMICAL-SYSTEM-BASED MACHINE

**(1) Hamiltonian and Natural Annealing:** DS-machines operate based on an energy function called the **Hamiltonian**, a key concept in physics that describes the total energy of a system. The Hamiltonian of the backbone DS-machine (Wu et al., 2024) is augmented from the Ising model which is a statistical physics model widely used in the modeling of interacting spins. The Hamiltonian defined as:

$$\mathcal{H} = -\sum_{i \neq j}^{N} J_{ij}\sigma_i\sigma_j + \sum_i^N h_i\sigma_i^2, \ \sigma_i \in \mathbb{R} \tag{1}$$

where $\sigma$ are system spins, $J$ are coupling parameters representing the correlations among spins, and $h$ are spins' self-reaction intensity to external influence. While retaining the strengths of the Ising model, this new Hamiltonian lifts its binary constraint to support real values, achieving high performance on graph learning problems.

The computing power of DS-machines stems from the process of **Natural Annealing**, an inherent characteristic of dynamical systems. In systems such as interacting spin models, the Hamiltonian naturally decreases due to spin interactions. From a physics perspective, this occurs because spins tend to settle into lower energy states, guiding the system toward optimal solutions. To harness Natural Annealing, the parameters $J$ and $h$ are programmed based on the target problem, shaping the Hamiltonian's energy landscape to align the desired outcomes with its minimum states. As the programmed DS-machine initiates Natural Annealing from random initial conditions, the system rapidly converges to an energy minimum, with the spin dynamics stabilizing at the target results.

**(2) DS-machine Hardware:** Since programming interacting spins can be prohibitively expensive, the backbone DS-machine (Wu et al., 2024) is built on electronic dynamical systems, which are implemented using current CMOS technology.

As Fig. 2 shows, corresponding to equation 1, each spin $\sigma_i$ is implemented as a nano-scale capacitor within a node unit ($N_i$), with its voltage representing spin value. Each capacitor is coupled with a programmable resistor, forming a feedback loop within the node unit that serves as the self-reaction parameters $h$. Capacitors from different node units ($N_i \& N_j$) are structurally connected by a programmable resistor in coupling unit $CU_{ij}$ to perform the coupling parameters $J$. Natural Annealing in this system is driven by voltage

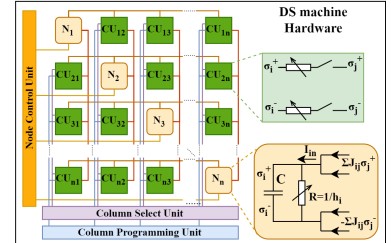

Figure 2: Backbone DS-machine.

imbalances across capacitors, which guide the natural movement of electrons toward equilibrium, propelling the system to evolve toward energy minima. The convergence of the Natural Annealing, in other words, the spontaneous energy decrease of the Hamiltonian with time ($d\mathcal{H}/dt \leq 0$) can be theoretically guaranteed by Lyapunov stability analysis (Blaquiere, 2012). As Fig. 2 shows, for each spin $\sigma_i$, its electrodynamics behavior is designed as:

$$\frac{d\sigma_i}{dt} = \sum_{i \neq j}^{N} J_{ij}\sigma_j - h_i\sigma_i = -\frac{1}{2}\frac{\partial \mathcal{H}}{\partial \sigma_i} \tag{2}$$

Then, following the chain rule we have:

$$\frac{d\mathcal{H}}{dt} = \sum_i \frac{\partial \mathcal{H}}{\partial \sigma_i}\frac{d\sigma_i}{dt} = -\frac{1}{2}\sum_i (\frac{\partial \mathcal{H}}{\partial \sigma_i})^2 \leq 0 \tag{3}$$

As the equations show, the system's electrodynamics inherently drive the Hamiltonian toward a local minimum. To escape local minima and achieve better solutions, several techniques can be employed such as spin-flipping and noise injection (Afoakwa et al., 2021).

### 2.2 LARGE LANGUAGE MODELS

In this paper, LLMs refer to pre-trained Transformer models such as GPT, Llama, Gemini, and Claude, which are among the most popular models today. Since the Transformer model was introduced in (Vaswani, 2017), it has achieved remarkable success across various domains, including natural language processing (NLP), computer vision (Wang et al., 2023), and audio processing (Ghosal et al., 2023). While modern LLMs are evolving towards multi-modal capabilities that integrate

diverse types of data, this study focuses on the classic Transformer architecture as an early-stage exploration.

A classic Transformer consists of both an encoder and a decoder, whereas mainstream commercial models like GPT are decoder-only. Despite this distinction, the key components of both encoder and decoder are similar: multi-head self-attention, feed-forward Multilayer Perceptrons (MLPs), linear projection layers, and layer normalization. This work primarily investigates decoder-based models, such as GPT-2 and OPT, while maintaining adaptability for other Transformer variants.

Although matrix multiplication remains a fundamental computational task in LLMs as in most neural networks, the primary computational bottleneck in LLMs is the attention layer. The multi-head self-attention mechanism, which significantly enhances LLM performance, imposes an exceptionally high computational burden. Considering each token $x_i$ as a row vector, the break down of multi-head self-attention of each head is:

$$q_i = x_i W_Q, \; k_i = x_i W_K, \; v_i = x_i W_V \tag{4}$$

$$f(q_i, k_j) = exp(\frac{q_i k_j^T}{\sqrt{d}}) \tag{5}$$

$$A_i = \sum_{j=1}^{N} \frac{f(q_i, k_j) v_j}{\sum_{j=1}^{N} f(q_i, k_j)} \tag{6}$$

where $W_Q$, $W_K$, and $W_V$ are learned weight matrices, and $q_i$, $k_i$, $v_i$ are the row vectors representing the *query*, *key*, and *value* for token $x_i$, respectively. Function $f(q_i, k_j)$ measures the similarity between query and key. Consequently, the computation in the attention layer involves weighted sum, matrix multiplications, and exponential function. Based on the above decomposition, the next chapter will demonstrate how the Transformer model can be implemented on DS-machines.

## 3 DS-LLM: MAPPING LLM ONTO DS-MACHINE

### 3.1 OVERVIEW

In this chapter, we provide a detailed, step-by-step guide on leveraging DS-machines to enhance both the inference and training of LLMs. As illustrated in Fig. 1, traditional transformer models are executed on digital GPUs using CUDA and tensor cores, whereas DS-LLM maps the original model onto the energy landscape of DS-machines, utilizing Natural Annealing for the forward pass. Furthermore, while conventional transformers rely on offline training via backpropagation, DS-LLM introduces an online **Electric-Current-Loss**-based training method to enable rapid back-propagation. By harnessing the immense computing power of DS-machines, DS-LLM accelerates both inference and training, offering a fundamentally more efficient approach to LLM optimization.

### 3.2 MAPPING METHOD

The mapping of existing models onto DS-machines can follow two distinct approaches: one at the Hamiltonian level and the other at the electrodynamics level. To illustrate these approaches, we begin with a single linear layer. Consider the input matrix $X \in \mathbb{R}^{n \times d_{in}}$, the output matrix $Y \in \mathbb{R}^{n \times d_{out}}$, and the weight matrix $W \in \mathbb{R}^{d_{out} \times d_{in}}$, along with an optional bias vector $b \in \mathbb{R}^{d_{out}}$. Since the bias term $b$ can be absorbed into the matrix multiplication operation, we omit it for simplicity. Consequently, the transformation performed by this layer can be expressed as a single matrix multiplication $Y = XW^T$.

**On the Hamiltonian level**, to leverage the Natural Annealing process on DS-machines, we must shape the energy landscape so that its minimum energy state corresponds to the desired output. To achieve this, we define a target function $F$ that minimizes the squared Frobenius norm of the difference (or Euclidean distance) between the output state of the DS-machine, $Y_{DS}$, and the desired output, $X_{DS}W^T$. This forms the following minimization problem:

$$F = \|Y_{DS} - X_{DS}W^T\|_F^2 = \sum_{i,l} (y_{il} - \sum_j w_{ij} x_{jl})^2 \tag{7}$$

where $w$, $x$, and $y$ represent elements of the matrices $W$, $X_{DS}$, and $Y_{DS}$, respectively, and $i$,$j$, and $l$ correspond to the dimensions $n$,$d_{in}$, and $d_{out}$. The target function $F$ reaches its minimum, $F_{min} = 0$, when $Y_{DS} = X_{DS}W^T$. Since the absolute value of $F_{min}$ is not crucial, and $\sum_j w_{ij} x_{jl}$ remains constant during inference, we can simplify the target function as:

Figure 3: Implementation of key operations.

$$\hat{F} = \sum_{(i,l)} y_{il}^2 - 2 \sum_{(i,l)} \left( y_{il} \sum_j^N w_{ij} x_{jl} \right) \tag{8}$$

After transforming the linear layer's computation into a minimization problem, we program the DS-machine by aligning its Hamiltonian with this simplified target function, $\hat{F}$. As shown in equation 8, $\hat{F}$ is a special case of the Hamiltonian described in equation 1. In this case, the spins $\sigma$ are divided into two groups: $x_{ij}$, representing the input, and $y_{il}$, representing the output. The self-reaction parameters $h_i$ are set to 1 for $y_{il}$, while the coupling parameters $J$ are assigned $2w_{ij}$ between corresponding spins $x_{jl}$ and $y_{il}$, with all other elements set to 0. The value of $h_i$ for $x_{ij}$ does not affect the convergence in equation 3 because the input $x_{ij}$ is fixed, resulting in $dx_{ij}/dt = 0$.

This configuration maps the target function $\hat{F}$ to the Hamiltonian of the DS-machine, enabling the computation of this layer via Natural Annealing process. During this process, the Hamiltonian will continuously decrease until it reaches a minimum (which is also the minimum of $\hat{F}$), where $Y_{\text{DS}} = X_{\text{DS}} W^T$ naturally emerges.

**On the electrodynamics level**, we can arrive at a similar conclusion. As seen in equation 2, the electrodynamics behavior is governed by the coupling parameters $J$ and self-reaction parameters $h$. According to Lyapunov stability analysis (Blaquiere, 2012), all spins should stabilize at a specific value when the system reaches a stable point, i.e., a local minimum. Hence, the electrodynamics of all spins must satisfy a boundary condition where $d\sigma/dt = 0$. Dividing the spins $\sigma$ into two groups, as before—input $x$ and output $y$ —along with the boundary condition, we arrive at:

$$y_i = \frac{\sum_j^N J_{ij} x_j}{h_i} \tag{9}$$

In this scenario, the spin electrodynamics exhibit a solvable stable point, enabling us to directly program $J$ and $h$ to map the desired matrix multiplication results onto the spin dynamics. Notably, this leads to the same mapping setup as derived from the Hamiltonian-level analysis.

It is a fortunate coincidence that the backbone DS-machine is inherently well-suited for matrix multiplication, a fundamental computational operation in LLMs. In the next subsection, we extend this mapping approach to more general functions, including other key operations in LLMs, where slight augmentations to the backbone DS-machine are required for full support.

### 3.3 TRANSFORMER IMPLEMENTATION

Recall the decomposition of the self-attention layer. Aside from the linear projection layer in equation 4, which has already been implemented, three key operations remain in the attention mechanism: (a) Query-Key matrix multiplication, (b) the exponential function, and (c) the weighted-sum operation. Operations (a) and (b) are defined in equation 5, while (c) is given in equation 6.

Fig. 3 illustrates the implementation of these three key operations, with detailed explanations provided below. For reference, the previously introduced implementation of the linear layer—responsible for matrix multiplication between weights and features—is also included in the figure.

**a) Query-Key Matrix Multiplication:** Unlike the typical multiplication in a linear layer, this operation occurs between two feature matrices generated online, rather than between features and weights. In the linear layer analysis, we assumed the weight matrix $W$ is obtained offline and loaded onto programmable resistors in the coupling units. Fortunately, these resistors in the backbone DS-machine are implemented using transistors, and programming them is accomplished by adjusting the voltage on their control ports—a common technique in CMOS design. Meanwhile, the spins in DS-machines are represented by capacitors, where the voltage corresponds to the spin

value. Thus, we can map the feature matrix $K$ onto the programmable resistors by connecting the output voltage of the spins to the control ports of the resistors with necessary scaling circuits.

**b) Exponential approximation:** The exponential function is computationally expensive and requires a high-order Taylor expansion for approximation. Based on a pre-experiment, we explore the trade-off between model accuracy and hardware cost and select a 3rd-order Taylor expansion as an approximation. Details of this trade-off can be found in Appendix.

$$exp(x_i)_{\text{Taylor3}} = 1 + x + 1/2x^2 + 1/6x^3 \tag{10}$$

As Fig. 3 shows, both the second and third order terms are implemented in the same way as the Query-Key Matrix Multiplication.

**c) Weighted-Sum:** Similar to the matrix multiplication, we can build the target function:

$$F = \sum_i (y_i - \frac{\sum_j f(q_i, k_j)v_j}{\sum_j f(q_i, k_j)})^2 = \sum_i (\frac{\sum_j f(q_i, k_j)y_i - \sum_j f(q_i, k_j)v_j}{\sum_j f(q_i, k_j)})^2 \tag{11}$$

Since the probability of $\sum_j f(q_i, k_j)$ keeping zero is negligible in an evolving dynamical system, we multiply it on equation 11 and reduce the constant terms:

$$\hat{F} = -\sum_{i,j} f(q_i, k_j)y_i v_j + \sum_i^N \frac{1}{2} \sum_j f(q_i, k_j)y_i^2 \tag{12}$$

In this setup, we program $f(q_i, k_j)$ to $J_{ij}$ like in the Query-Key Matrix Multiplication. Notice that $\frac{1}{2}\sum_j f(q_i, k_j)$ can be regarded as a matrix multiplication operation between $f(q_i, k_j)$ and an all-ones vector, which can be mapped to DS-machine as a simplified linear layer. The results are connected to the programmable resistors in node units, where $h_i = \frac{1}{2}\sum_j f(q_i, k_j)$ for $y_i$, and set $h_i = 0$ for $v_j$. With this setup, the Weighted-Sum operation in equation 6 can also be mapped to the Hamiltonian described in equation 1.

Now we have most of the essential components of a transformer model. The other operations which have relatively lower computing demanding like LayerNorm and activation are handled by auxiliary functional units. Details of other operations can be found in the appendix. Next, we demonstrate how to map the entire transformer model onto a DS-machine. Assuming the original model consists of multiple layers, each decomposed into $P$ operations, its function can be expressed as:

$$y = f^{(P)} \circ f^{(P-1)} \circ \cdots \circ f^{(2)} \circ f^{(1)}(x) \tag{13}$$

Since we have confirmed that the mapping method works for each computational component, we can now construct the general target function as follows:

$$\hat{F}^{(p)} = (x^{(p+1)})^2 - 2x^{(p+1)}f^{(p)}(x^{(p)}) \tag{14}$$

Here, we combine the mapping of all individual target functions by using the output of each lower layer as the input to the higher layer, where $x^1$ is the initial input and $x^{P+1}$ is the final output. It's important to note that when combining them, the influence between spins is unidirectional—from lower to higher layers—i.e., $\partial \hat{F}^{(p+1)}/\partial x^{(p)} = 0$. Therefore, the Natural Annealing process converges in each operation, ensuring that $\hat{F}^{(P)}$ reaches its minimum when $x^{(P+1)} = f^{(P)} \circ f^{(P-1)} \circ \cdots \circ f^{(1)}(x^1)$.

In this setup, the entire transformer-based model can be mapped onto DS-machines. With global natural annealing, the model achieves the desired output when the system reaches its global energy minimum.

### 3.4 ONLINE TRAINING WITH ELECTRIC-CURRENT-LOSS

After accelerating the forward pass of LLMs using Natural Annealing, we further enhance their training. The core idea is that a well-trained DS-machine should reach its energy minimum when its output spins align with the ground truth from the training data. Building on our electrodynamics analysis of the mapping method, we introduce an Electric-Current-Loss-based training method,

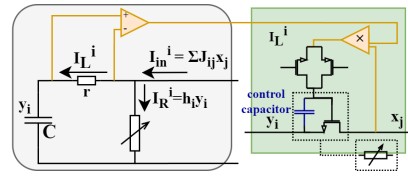

Figure 4: The feedback path of Electric-Current-Loss-based training.

which leverages the physical currents in the DS-machine to enable rapid online training.

As shown in Fig. 4, when Natural Annealing converges, for each output spin $y_i$, the total incoming electric current $I_{in}^i = \sum_j^N J_{ij} x_j$ at the node unit must balance the internal current $I_R^i = h_i y_i$ flowing through the resistor within the same node, ensuring that the capacitor voltage (representing the spin values) remains stable ($dy_i/dt = 0$). Referring to equation 9, if we map the ground truth output $\hat{y}_i$ onto the output spins and fix their values, we can define the internal current reference as $\hat{I}_R^i = h_i \hat{y}_i$. The difference between the incoming current $I_{in}^i$ and the reference current $\hat{I}_R^i$ forms a new loss function, expressed as $L_{EC} = \sum_i^N (I_{in}^i - \hat{I}_R^i)^2$. This difference is equal to the current through the sampling resistor $r$, $I_L^i = I_{in}^i - \hat{I}_R^i$. We then update $J_{ij}$ ($h_i$ = 1 is constant) using gradient descent:

$$\Delta J_{ij} = \frac{\partial L_{EC}}{\partial J_{ij}} = \frac{\partial L_{EC}}{\partial I_L^i} \frac{\partial I_L^i}{\partial J_{ij}} = 2I_L^i x_j \tag{15}$$

As depicted in Fig. 4, the multiplication of the loss current $I_L^i$ by the spin values $x_j$ or $\hat{y}_i$ can be implemented at the circuit level. The resulting current is then fed back to the control port of the programmable resistors in the coupling units (CUs) and nodes. Consequently, the parameters are updated as $J_{ij} \rightarrow J_{ij} - \Delta J_{ij} \Delta t$ by integrating the result current on the control capacitor of the programmable resistors over a time interval $\Delta t$.

We propose this training method, named Electric-Current-Loss (ECL) online training, which aims to minimize the loss function $L_{EC}$. Currently, this method only supports single-layer DS-machines, where a ground truth $\hat{y}_i$ is available. In this early-stage research, we leverage conventional back-propagation training to provide a baseline on how to combine ECL with backpropagation to enable efficient LLM training on DS-machines.

In the training of most LLMs, the loss in the output layer is typically computed using softmax with cross-entropy, yielding $\partial L / \partial x_i = y_i - \hat{y}_i$, where $x_i$ represents the input logits. Referring to equation 9 and equation 14, after the DS-machine completes the Natural Annealing process, the output spin value $y_i$ should converge to the desired computational result $f^{(P)}(x^{(P)})$. If we map the ground truth $\hat{y}_i$ to the output spin of the final layer, we obtain:

$$\frac{\partial L}{\partial x_i} = y_i - \hat{y}_i = \frac{I_{in}^i - I_R^i}{h_i} = I_L^i \tag{16}$$

Thus, using the ECL training method, the gradients of the logits can be represented as an electric current. As shown in Fig. 5, the gradients in other layers are computed based on the activations and the electric current loss from the subsequent layer. Since the gradient of a matrix multiplication operation is itself a matrix multiplication, this computation can also be efficiently mapped onto DS-machines. For certain non-polynomial operations, additional auxiliary circuits handle the gradient calculation.

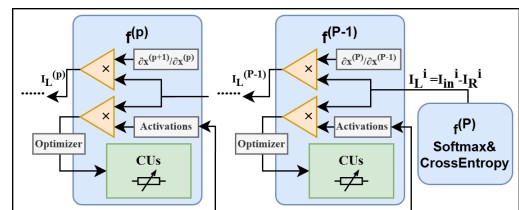

Figure 5: ECL-based backpropagation.

Moreover, because the forward mapping process is directional, the connections and computational units (CUs) from later layers to earlier layers remain unused during inference. We exploit these idle CUs for gradient computation without incurring additional cost. The gradients are then integrated on the controlling capacitors, updating the parameters accordingly. In this way, we seamlessly combine ECL with backpropagation, enabling an online training method that eliminates output readout while leveraging the fast forward computation of DS-machines.

## 4 EVALUATION

### 4.1 MODEL AND TRAINING SETUP

**Models, Tasks and Datasets:** We apply DS-LLM to several open-source models: GPT-2 (base and medium) (Radford et al., 2019), OPT-1.3B and OPT-2.7B (Zhang et al., 2022), and Llama2-7B(Touvron et al., 2023). For all models, we fine-tune and evaluate on five datasets from the GLUE

benchmark (Wang et al., 2019): SST-2 (Socher et al., 2013) for Single-Sentence Tasks, MRPC (Dolan & Brockett, 2005) and QQP (DataCanary et al., 2017) for Paraphrase Tasks, QNLI (Rajpurkar et al., 2016) and RTE (Dagan et al., 2006) for Inference Tasks. Additionally, we pre-train GPT-2-medium from scratch on OWT (Gokaslan & Cohen, 2019).

**Experiments Setup:** The fine-tuning on the GLUE benchmark was conducted on 4 Nvidia A100 40GB GPUs. The global batch size was set to 32 for GPT-2 (124M), 16 for GPT-2-medium (355M) and OPT-1.3B, and 8 for OPT-2.7B and Llama2-7B. We fine-tuned all models for 2 epochs on QQP and 3 epochs on the other datasets. The optimizer used was AdamW with an initial learning rate of 2e-5, and all other parameters were kept as default, as provided by the Hugging Face Transformers library. For pre-training GPT-2-medium, we utilized 80 Nvidia A100 40GB GPUs with a global batch size of 480 and trained the model for 120,000 iterations. The optimizer was AdamW with a 6e-4 initial learning rate, and other parameters were also kept at default settings. The inference experiments are using the same global batch size as in the fine-tuning experiments. For DS-Machine evaluation, the Natural Annealing process is evaluated with a standard Finite Element Analysis (FEA) software emulator. Additionally, the power consumption of DS-Machine is generated using the Cadence Mixed-Signal Design Environment with 45 nm CMOS technology.

## 4.2 ACCURACY COMPARISON BETWEEN DS-LLM AND ORIGINAL LLMs

In order to verify the accuracy loss of the proposed mapping method (DS-LLM) and training method (ECL), we fine-tune and evaluate the models on selected datasets to compare the accuracy across three model types: a) the original LLMs, trained offline and inferred on GPUs; b) DS-LLM models, trained offline on GPUs but inferred on DS-machines; and c) DS-LLM-ECL models, trained online and inferred on DS-machines. As shown in Table 1, the accuracy of DS-LLM models is comparable to that of the original LLMs, with some even outperforming them, demonstrating that the approximation loss during mapping is minimal and the mapping method is effective. The DS-LLM-ECL models also maintains good accuracy, validating the feasibility of the ECL online training method.

Table 1: Accuracy comparison (in Accuracy (%)): the higher the better.

| Task | Paraphrase Tasks | | Inference Tasks | | Single-Sentence Tasks |
|---|---|---|---|---|---|
| **Dataset** | MRPC | QQP | RTE | QNLI | SST2 |
| **GPT2** | 75.00 | 88.43 | 63.18 | 88.03 | 91.63 |
| **GPT2-DS** | 76.72 | 89.04 | 63.54 | 88.28 | 91.29 |
| **GPT2-DS-ECL** | 76.91 | 89.24 | 63.11 | 87.94 | 90.82 |
| **GPT2-M** | 79.66 | 90.57 | 68.59 | 91.05 | 93.58 |
| **GPT2-M-DS** | 79.17 | 90.55 | 70.40 | 90.33 | 93.35 |
| **GPT2-M-DS-ECL** | 79.31 | 90.09 | 70.41 | 89.73 | 93.06 |
| **OPT1.3B** | 84.07 | 90.94 | 77.26 | 91.09 | 92.43 |
| **OPT1.3B-DS** | 87.01 | 88.48 | 78.70 | 91.41 | 92.20 |
| **OPT1.3B-DS-ECL** | 86.57 | 88.59 | 78.05 | 91.45 | 91.81 |
| **OPT2.7B** | 86.52 | 91.03 | 82.33 | 93.15 | 94.08 |
| **OPT2.7B-DS** | 86.54 | 90.98 | 82.48 | 93.27 | 94.04 |
| **OPT2.7B-DS-ECL** | 86.74 | 90.67 | 82.44 | 93.41 | 94.25 |
| **Llama2-7B** | 90.01 | 91.10 | 88.45 | 95.75 | 96.58 |
| **Llama2-7B-DS** | 89.47 | 90.95 | 88.70 | 95.69 | 96.32 |
| **Llama2-7B-DS-ECL** | 89.56 | 91.08 | 88.57 | 95.73 | 95.79 |

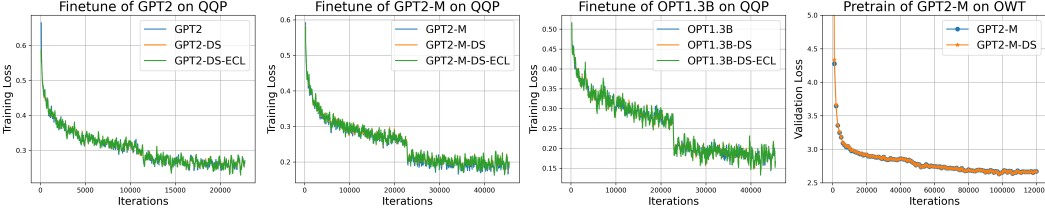

Figure 6: Visualization of training trajectories.

The fine-tuning trajectories of the original LLMs, DS-LLM, and DS-LLM-ECL models are visualized in Fig. 6. Additionally, we present the training curves of the original LLMs and DS-LLM during GPT-2 Medium pretraining; unfortunately, we are unable to pretrain DS-LLM-ECL models due to computational limitations. Notably, both DS-LLM and DS-LLM-ECL exhibit convergence curves that closely match those of the original models, demonstrating that the mapping and online training methods effectively replicate the performance of traditional LLMs on DS-machines.

Table 2: Performance comparison between GPU and DS-machines.

| Metric | Training | | Inference | | |
|---|---|---|---|---|---|
| | Throughput (token/s) | Energy Efficiency (token/KWh) | Time to First Token (s) | Token Generation Rate (token/s) | Energy Efficiency (token/KWh) |
| GPT2 | 1.37E+04 | 6.19E+07 | 6.46E-05 | 1.55E+04 | 1.39E+08 |
| GPT2-DS | 6.70E+06 | 8.93E+12 | 1.20E-06 | 8.33E+05 | 1.11E+12 |
| gpt2-m | 3.24E+03 | 1.46E+07 | 1.80E-04 | 5.55E+03 | 5.18E+07 |
| gpt2-m-DS | 3.27E+06 | 4.36E+12 | 2.48E-06 | 4.03E+05 | 5.38E+11 |
| OPT1.3B | 1.20E+03 | 5.41E+06 | 2.76E-04 | 3.62E+03 | 3.41E+07 |
| OPT1.3B-DS | 3.08E+06 | 9.56E+11 | 2.55E-06 | 3.92E+05 | 1.22E+11 |
| OPT2.7B | 3.97E+02 | 1.78E+06 | 8.12E-04 | 1.23E+03 | 1.13E+07 |
| OPT2.7B-DS | 2.33E+06 | 4.37E+11 | 3.41E-06 | 2.93E+05 | 5.50E+10 |
| Llama2-7B | 1.34E+02 | 6.05E+05 | 2.51E-03 | 3.98E+02 | 3.82E+06 |
| Llama2-7B-DS | 2.24E+06 | 1.56E+11 | 3.57E-06 | 2.80E+05 | 1.95E+10 |

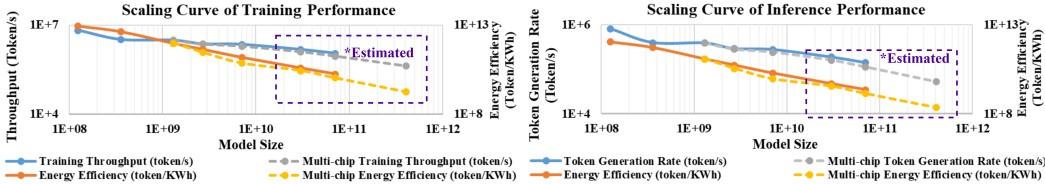

Figure 7: Scaling curves of single-chip and multi-chip solutions.

## 4.3 PERFORMANCE COMPARISON

Table. 2 compares the average training and inference performance on all dataset and estimated energy consumption between the implementation on GPUs and our solution. The power consumption of the 4 A100 GPUs is estimated at 800 watts, and the inference evaluation on GPUs is based on vLLM(Kwon et al., 2023) for fair comparison.

As shown in Fig.7, we visualized the scaling trends of DS-LLM to show how the performance change with model size, illustrating linear reductions in token rate and energy efficiency as model size increases. The models larger than 7B are estimated by linearly extrapolation from 7B. There are two scaling solutions shown in the figure, including single-chip solution which is used in Table. 2 and multi-chip solution which we suggest to handle larger models. The motivation of a multi-chip solution

Table 3: Hardware cost with single-chip scaling.

| Model size | Area (mm$^2$) | Power (W) |
|---|---|---|
| GPT2-DS | 19.8 | 2.7 |
| gpt2-m-DS | 19.8 | 2.7 |
| OPT1.3B-DS | 83.2 | 11.6 |
| OPT2.7B-DS | 131.1 | 19.2 |
| Llama2-7B-DS | 332.4 | 51.6 |
| OPT30B-DS (est.) | 907.5 | 138.2 |
| OPT66B-DS (est.) | 1448.6 | 226.7 |

comes from the considerations on hardware implementation. As we demonstrate in Table. 3, the chip area and power consumption increase quickly with model size under a single-chip solution and the area needed for a 66B model is even larger than commercial GPUs. Building extremely big single chip will increase the practical risks due to physical limitations such as process yield and thermal management. Though the industry has built some wafer-scale extremely large chips (Hu et al., 2024), multi-chip scaling out solution is a more mature and widely used solution in AI accelerator industry. However, while multi-chip solution involves fixed chip sizes, additional communication overhead will reduce the system efficiency as shown in the figure. The performance of actual multi-chip systems will also be influenced by the implementation of memory hierarchy, communication bandwidth, and many system level trade-offs.

For a more comprehensive evaluation on inference, we also compared our DS-LLM with some low-precision implementations for edge devices, including low-precision CPU(Shen et al., 2023) and current SOTA accelerator for LLM, Cambricon-LLM(Yu et al., 2024). As shown in Table. 4, DS-LLM outperforms the references on both token rate and energy efficiency. Results of Cambricon-LLM are sourced from the original paper.

Table 4: Comparison on Llama2-7B with low-precision CPU and edge devices.

| Solutions | Token Generation Rate (tokens/s) | Energy Efficiency (tokens/KWh) |
|---|---|---|
| Low Precision CPUs(Shen et al., 2023) | 45.4 | 1.63E+6 |
| Cambricon-LLM(Yu et al., 2024) | 3.55 | 3.60E+6 |
| DS-LLM (this work) | 3.03E+4 | 4.04E+10 |

We emphasize that our assessment follows standard analog simulation flows, while acknowledging that it may not fully capture the complexities of real physical chip measurements. Nonetheless, we

believe the demonstrated orders-of-magnitude improvements in speed and energy efficiency highlight the potential of DS-LLM as a promising solution for overcoming bottlenecks in LLM computation.

## 5 DISCUSSION

While DS-machines demonstrate significant theoretical potential as an emerging solution, this section explores the practical challenges they may face, aiming to lay a solid foundation for future research and development in this promising field. We address key aspects such as scalability, model flexibility, physical practicality, system integration feasibility, robustness, and precision constraints, supported by preliminary verification. Due to space limitations, some discussions are included in the Appendix.

**Scalability:** While prior work like NP-GL was designed for small graphs with fewer than 1,000 nodes, the capacity of DS-machines can be significantly expanded to handle much larger scales. First, DS-machines have demonstrated linear complexity with respect to the number of nodes (Song et al., 2024), making them inherently scalable with increased chip area. For context, NP-GL occupies only about 5 $mm^2$, whereas modern GPUs like the H100 have a die size of 814 $mm^2$, and wafer-scale chips—such as those with up to 46,255 $mm^2$—are emerging (Hu et al., 2024). Based on linear complexity, a single-chip solution could theoretically support millions of nodes within a single DS-machine. Second, for even larger models or faster training, multi-chip approaches offer a viable path forward. Existing research has explored multi-chip solutions for DS-machines (Sharma et al., 2022), where individual chips perform annealing with periodic synchronization. We are also investigating promising techniques such as deploying models across multiple DS-machine chips to achieve pipeline parallelism. Overall, the scalability of DS-machines is theoretically well-founded, offering both single-chip and multi-chip solutions to meet the demands of increasingly large and complex models.

**Model Flexibility:** This work focuses on classic operations in Transformer-based LLMs, recognizing that modern LLMs may incorporate different operations, such as diverse activation functions or embedding methods. Despite the variety of LLM architectures, these operations can generally be categorized as either polynomial or non-polynomial. Polynomial operations, which can be decomposed into basic addition and multiplication, are directly mappable to DS-machines. Non-polynomial operations, such as the exponential function, require either transformation into polynomial approximations (e.g., via Taylor expansion) or the addition of auxiliary circuits, which may slightly increase latency depending on their complexity. Fortunately, most high-computational-demand operations, particularly those in attention layers and feed-forward MLPs, are polynomial or even linear. Thus, DS-machines offer significant flexibility and adaptability for a wide range of models.

## 6 RELATED WORK

The early-stage research on DS-machines is rooted in the Ising model, which supports only binary spin values and primarily addresses binary optimization problems (Afoakwa et al., 2021). Our backbone DS-machine was proposed in NPGL (Wu et al., 2024) and applied to graph learning problems. However, NPGL employs an individual learning method that ignores the architecture of Neural Networks, limiting its ability to leverage existing technologies effectively.

There are several variants of dynamical systems, such as optical (Inagaki et al., 2016) and oscillator-based (Lo et al., 2023) Ising machines. Although these approaches show promise, they have yet to be successfully integrated into machine learning applications. To the best of our knowledge, this work represents the first attempt to combine existing machine learning models, particularly Large Language Models (LLMs), with DS-machines.

## 7 CONCLUSION

In this work, we introduce DS-LLM, the first algorithmic framework that bridges LLMs to existing DS-machines, harnessing the power of Natural Annealing to efficiently execute LLMs on DS hardware. The mathematical equivalence between DS-LLM and original LLMs is proven and validated through experiments on models from GPT-2 to Llama2-7B. Results demonstrate consistent accuracy while achieving orders-of-magnitudes speedup and energy reduction on both training and inference. In conclusion, DS-LLM presents a promising new solution for the community with significant opportunities for further exploration in future studies.

ACKNOWLEDGEMENTS

This work is supported by the U.S. Department of Energy, Office of Science, Office of Advanced Scientific Computing Research, in support of the MEERCAT Microelectronics Science Research Center, under Contract DE-AC05-76RL01830. This work is also supported by NSF under Award No. 2326494.

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

## A APPENDIX

### A.1 DISCUSSION

As an early-stage research initiative, DS-LLM aims to introduce a promising new computing paradigm to meet the growing computational demands of LLMs. While still in its infancy, this section discusses and analyzes its potential, challenges, and future directions.

**a) System Integration Feasibility:** First, we emphasize that there are no fundamental challenges to integrating DS-machines into existing computing systems. Although architecturally distinct from digital processors, DS-machines are built using CMOS-compatible technology, ensuring they can be seamlessly integrated as co-processors (similar to TPUs or NPUs) via interfaces like PCIe. No major hardware adaptations are theoretically required. From a system perspective, while this work is still in its early stages, we believe the integration of DS-machines into existing computing infrastructures holds great promise for future exploration. There is significant potential in developing software toolchains, such as compilers, optimizing memory management, and pipelining tasks between DS-machines and other processors. With these advancements and supporting software tools, we are confident that DS-machines are inherently feasible for integration into digital systems as a new type of co-processor, akin to GPUs, TPUs, or NPUs. Future work could explore hybrid use cases that combine CPUs, GPUs, and DS-machines, leveraging their unique strengths to achieve optimal performance. Overall, while DS-machines are not yet fully mature, their fundamental feasibility opens the door to exciting opportunities for integration into existing computing infrastructures.

**b) Robustness on Circuit Non-idealities:** For a comprehensive discussion, we address two types of Non-idealities: dynamic noise (e.g., thermal noise) and static offset (e.g., fabricated non-linearity and mismatch).

**Dynamic noise:** It has been proved that stochastic process like the natural annealing in DS-machines is usually more robust to noise than deterministic process(Ohayon et al., 2023). We setup an experiment to evaluate the impact of varying noise levels on DS-LLM, where two models are fine-tuned

Table 5: Influence of dynamic noise on accuracy of DS-LLM.

| Model | Noise Level | MRPC | QQP | RTE | QNLI | SST2 |
|---|---|---|---|---|---|---|
| gpt2-DS-ECL | 0 | 76.91 | 89.24 | 63.11 | 87.94 | 90.82 |
| gpt2-DS-ECL | 0.05 | 76.84 | 89.20 | 63.02 | 87.83 | 90.77 |
| gpt2-DS-ECL | 0.10 | 76.55 | 88.93 | 62.71 | 87.56 | 90.33 |
| gpt2-DS-ECL | 0.15 | 75.89 | 88.12 | 62.05 | 86.98 | 89.84 |
| OPT1.3B-DS-ECL | 0 | 86.57 | 88.59 | 78.05 | 91.45 | 91.81 |
| OPT1.3B-DS-ECL | 0.05 | 86.43 | 88.52 | 77.99 | 91.40 | 91.75 |
| OPT1.3B-DS-ECL | 0.10 | 86.15 | 88.31 | 77.63 | 91.22 | 91.43 |
| OPT1.3B-DS-ECL | 0.15 | 85.67 | 87.75 | 77.24 | 90.89 | 91.01 |

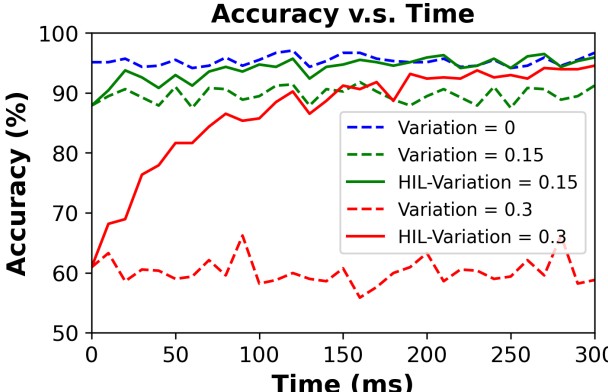

Figure 8: Accuracy recovery for a pre-trained GPT2 model during finetuning on DS-machines.

on the downstream datasets based on our work with dynamic noise injected into the simulation. The dynamic noise was modeled as standard Gaussian noise with a standard deviation ranging from 0.05 to 0.15. The results demonstrate that DS-machines exhibit high robustness to dynamic noise.

**Static offset:** Static offset is a common challenge in analog circuits, arising from factors like hardware non-linearity and mismatches during fabrication. Our proposed online training method addresses this effectively by training and performing inference on the same hardware device. This ensures that the model inherently adapts to the hardware's biased patterns during training, resulting in accurate inference despite non-idealities. Moreover, when deploying a pre-trained model on different devices, a promising solution is to fine-tune the model on the target hardware for a few batches. This allows the model to quickly adapt to the specific biased pattern of the new hardware. We evaluate this method by finetuning a pre-trained GPT2 model on DS-machines with different offset level. The offset of DS-machines is modeling as standard gaussian noise with standard variation from 0 to 0.3. As shown in Fig. 8, the dash line represents the inference accuracy without finetuning, while the solid line represents the inference accuracy with finetuning. The results illustrates the recovery of model accuracy during the fine-tuning process, underscoring the practicality and effectiveness of this approach.

**c) Precision Constraints:** The precision of computation in DS-machines is inherently continuous due to their analog nature. However, the precision of Analog-to-Digital Converters (ADCs) and Digital-to-Analog Converters (DACs) used in the system do affect the overall accuracy. Fortunately, ADCs and DACs are well-established technologies with numerous mature solutions that allow for various design trade-offs—for instance, achieving 16-bit precision with lower power consumption or 32-bit precision with higher power consumption. Such trade-offs align with and are theoretically compatible with existing quantization techniques, providing flexibility to balance precision and power efficiency based on the application requirements.

**d) Advantages over other emerging computing diagram:** DS-machines show high potential to satisfy the increasing LLM computing demanding, while there are also other emerging computing diagrams like quantum computing, optical computing and Computing-In-Memory works. We would like to briefly compare DS-machines with other approach and highlight our key advantages.

**Quantum computing:** Quantum computing is a promising avenue but is still constrained by scalability issues and the need for complex error correction. As quantum systems scale, errors and noise increase, demanding advanced error-correction techniques that are not yet fully mature. Existing largest quantum computer from IBM has about 1100 qubits, which is too small to support LLM computing tasks. Meanwhile, due to the technology requirements, nowadays building and running quantum computers are still very expensive. In the contrast, DS-machines are built on CMOS-compatible technologies, which are highly mature and the cost of its fabrication is at the same level of digital processors.

**Optical Computing:** The optical computing solutions also rely on special technology and is hard to be integrated with digital systems. Building complex and accurate optical circuits can be a big challenge and very expensive, making the stability and feasibility of optical computer a larger challenge than DS-machines.

**Computing-In-Memory (CIM)**: CIM technology is relatively feasible and CMOS compatible. Existing CIM works only supports inference, achieving up to 100s times acceleration and 2000 times energy reduction (Wolters et al., 2024), which is much lower than our solution especially on energy reduction. The insight behind this is that CIM still follows a traditional instruction-based paradigm, completing computations step by step. In contrast, DS-machines are driven by physical processes, specifically natural annealing, which doesn't require step-by-step control. This allows DS-machines to naturally perform complex tasks automatically without extra energy for controlling, achieving a much higher energy efficiency.

## A.2   TRADE-OFF ON TAYLOR EXPANSION

Table 6: Trade-off on Taylor Expansion order.

| Taylor Expansion order | Validation loss | Loss Drop | Hardware cost (units) |
|---|---|---|---|
| Inf (baseline exponential) | 3.25 | 0 | - |
| 1 | 3.35 | 0.10 | 1 |
| 3 | 3.28 | 0.03 | 6 |
| 5 | 3.27 | 0.02 | 15 |
| 7 | 3.27 | 0.02 | 28 |

Before training the models, we conducted a pre-experiment by fine-tuning a pre-trained GPT-2 model on a small dataset (Shakespeare, 300k) with different orders of Taylor expansion. The results show that the 3rd-order expansion achieves sufficiently low validation loss, while the improvement from higher orders is marginal. In terms of hardware requirements, the resource cost scales approximately linearly with the order of each term. For instance, a polynomial like "$x^3 + x^2 + x$" requires around 6 resource units, while adding a term like "$x^5$" would demand an additional 5 units. Based on this trade-off, we selected the 3rd-order expansion as the most balanced design choice. For those prioritizing accuracy over hardware efficiency, higher-order expansions can be adopted and are compatible with our framework.

## A.3   IMPLEMENTATION OF OTHER OPERATIONS

Due to the page limit, we only introduce the most important computing demanding operations in Section 3. Here we provide the analysis and implementation of other operations.

**a) Activation Functions:** The activation function used in GPT-2 at its initial publication was the ReLU function. However, many large language models (LLMs) have since transitioned to the Gaussian Error Linear Unit (GELU) (Hendrycks & Gimpel, 2017) for enhanced performance.

The ReLU function is straightforward to implement in hardware by simply turning off the output for spins with negative values. In contrast, the GELU function is more complex and is defined as follows:

$$\text{GELU}(x) = 0.5x \left( 1 + \tanh \left( \frac{\sqrt{2/\pi}(x + 0.044715x^3)}{2} \right) \right) \tag{17}$$

The GELU function can be decomposed into matrix multiplication operations and the hyperbolic tangent function. To approximate the tanh function, we employ the same Taylor series expansion method used for the exponential function:

$$\tanh_{\text{Taylor}}(x) = x - \frac{x^3}{3} + \frac{x^5}{5} \tag{18}$$

Consequently, the implementation of the GELU function can be transformed into a series of matrix multiplication operations, similar to the approach introduced for the exponential function in Fig. 3 in Section 3.3.

**b) Layer Normalization:** Layer normalization is a crucial function in Large Language Models (LLMs). Unlike traditional Convolutional Neural Networks (CNNs), where normalization is performed in the batch direction and can be fixed during inference, layer normalization involves complex computations that cannot be avoided:

$$\mu = \frac{1}{H} \sum_{i=1}^{H} x_i \tag{19}$$

$$\sigma^2 = \frac{1}{H} \sum_{i=1}^{H} (x_i - \mu)^2 \tag{20}$$

$$\hat{x} = \frac{x - \mu}{\sqrt{\sigma^2 + \epsilon}} \tag{21}$$

$$y = \gamma \hat{x} + \beta \tag{22}$$

In this formulation, we can observe that the calculations of $\mu$ and $y$ can be directly mapped to a series of matrix multiplication operations, as previously discussed. Additionally, reduction circuits, such as differential amplifiers, must be incorporated to handle the difference between $x_i$ and $\mu$. Furthermore, computing $\hat{x}$ requires an additional circuit to manage the division operation. Thus, the entire layer normalization operation can be effectively mapped to DS-machines.

