# OpenReview forum: "DS-LLM: Leveraging Dynamical Systems to Enhance Both Training and Inference of Large Language Models"
_ICLR.cc/2025/Conference — ICLR 2025 Poster_

### Official Review · Reviewer_HAWr · 2024-10-29

**Soundness:** 2
**Presentation:** 3
**Contribution:** 2
**Rating:** 6
**Confidence:** 2

**Summary:**

This paper proposes a novel approach to training and inference for large language models on dynamic systems, a new type of computing mechanism. Inference is facilitated through Hamiltonian configurations of the dynamic system machines, whereas training necessitates additional analog hardware for gradient computations. The authors assert that their approach yields substantial benefits in terms of speedup and energy reduction compared to traditional GPU-based methods.

**Strengths:**

(1) New computing paradigm. The authors explore the use of DS machines, a novel computing paradigm for both training and inference of large language models, a timely contribution given the substantial resources currently consumed by LLM pretraining.

(2) Significant gains. The authors claim 1000x speedups and 100,000x energy reduction for training while maintaining consistent model accuracy.

(3) Mathematically rigorous. Although the reviewer may not have understood every detail, the problem formulation of LLM inference into Hamiltonian and Natural Annealing in dynamical systems appears mathematically solid and promising.

**Weaknesses:**

(1) **Lack of clear differentiation from prior work**: One of the major contributions of using dynamical systems for machine learning appears to be extremely similar to prior work [1]. The reviewer struggles to understand the significant differences and contributions of this work compared to [1], particularly given that some graph neural networks in [1] also employ attention mechanisms. A clearer explanation of the novelty and significance of this work is necessary.

(2) **Inadequate consideration of circuit non-idealities**: The training process relies on specialized analog circuits to achieve gradient calculations and backpropagation. However, analog circuits are prone to non-idealities such as thermal noise and non-linearity [2]. The authors seem to overlook these non-idealities and their potential impact on model accuracy. Furthermore, the exponential approximation using Taylor expansion in Section 3.3 lacks justification, and the choice of a 3rd-order expansion is not fully explained. The authors should provide more insight into how this choice affects model accuracy and hardware requirements.

(3) **Limited analysis of model scalability**: The authors conduct analysis on model sizes of 124M, 355M, and 1.3B, but recent works have demonstrated the importance of scaling up model sizes, such as the 405B model in llama3. Can the authors provide more theoretical analysis or insight into how the speedups and energy consumption scale with the size of the models?

(4) **Unfair/Unclear comparisons**: The authors compare their energy calculations to A100 GPUs (800W), but the basis for the claimed 1000x speedup is unclear. Specifically, did the authors consider the latest techniques for fully utilizing GPU resources, such as vLLM[3]? Moreover, was batched inference used, considering that only 1.3B models were tested? It is unclear how/why this was not done, as four 40GB A100s would not be required for 124M models, for example. Especially in terms of LLM inference benchmarking, what is the comparision results on meaningful metrics, such as Time to First Token (TTFT),Token Generation Rate , and Energy in Token/KWh [4,5].?


References

[1] EXTENDING POWER OF NATURE FROM BINARY TO REAL-VALUED GRAPH LEARNING IN REAL WORLD

[2] Design of Analog CMOS Integrated Circuits

[3] Efficient Memory Management for Large Language Model Serving with PagedAttention

[4] https://www.anyscale.com/blog/comparing-llm-performance-introducing-the-open-source-leaderboard-for-llm

[5] https://huggingface.co/spaces/optimum/llm-perf-leaderboard

**Questions:**

**Regarding novelty and differentiation from prior work**

1. Can you provide a detailed comparison of your work with [1], highlighting the specific contributions and differences in your approach?
How does your use of dynamical systems for machine learning improve upon or depart from existing methods, particularly those that also employ attention mechanisms?
2. What are the key advantages of your approach over prior work (and other emerging compute alternatives such as compute-in-memory), and how do you demonstrate these advantages in your experiments?

**Regarding circuit non-idealities**

1. How do you plan to address the potential impact of thermal noise and non-linearity in analog circuits on model accuracy?

2. Can you provide a justification for the exponential approximation using Taylor expansion in Section 3.3, and explain why a 3rd-order expansion was chosen?How does the choice of Taylor expansion order affect model accuracy and hardware requirements, and can you provide experimental or theoretical evidence to support your choice?

**Regarding model scalability**

1. Can you provide a theoretical analysis of how your approach scales with increasing model sizes, such as those demonstrated in recent works (e.g., 405B model in llama3)?How do you expect the speedups and energy consumption to change as model sizes increase, and what are the implications for hardware requirements? Can you provide additional experiments or simulations to demonstrate the scalability of your approach?

**Regarding comparisons and metrics**

1. Can you clarify the basis for the claimed 1000x speedup, and provide more details on the experimental setup and metrics used?

2. Did you consider using the latest techniques for fully utilizing GPU resources, such as vLLM[3], and if not, why not? Why was batched inference not used, particularly for smaller models (e.g., 124M), and how would this affect the energy calculations and comparisons with A100 GPUs?

3. In terms of LLM inference/training benchmarking, what is the comparision results on meaningful metrics, such as Time to First Token (TTFT), Token Generation Rate, and Energy (Token/KWh).

4. Can the authors also compare with (to a rough level) on dynamical systems with other emerging computing candidates (as introduced in the second paragraph of the introduction), such as quantum computing, optical computing, computing in memory etc?

**Minor recommendations**: The abstract of the paper consists of 3 paragraphs and is quite long, can the authors abbreviate the abstract into a single paragraph to breifly explain the papers novelty and contributions?

---

> ### Author Response · Authors · 2024-11-21
> **Part1 of Response to Reviewer HAWr**
>
> ## Overview
> We sincerely appreciate for your insightful and constructive feedback, and we have uploaded a revised version of paper to address your concerns. The contents corresponding to your questions and concerns are labled as Reviewer D like "D-Q1-1". The detailed response to your comments are listed as below:
>
> ## 1. Novelty and Differentiation from prior work (question 1-1):
> Thank you very much for your meaningful comments, and we have revised the introduction to clearly highlight the distinctions between our work and the prior NPGL approach. We also add a discussion section in Appendix to clarify our novelty and differentiation from prior work, which is also listed as below.
>
> The prior work NPGL is not a mapping of existing GNNs including those who have attention mechanism. It ignores the architecture of neural networks and directly fits the data with the Hamiltonian, which is a shallow fully connected model without any hierarchy or structure as shown in Eq. (1). The parameter size of this model is determined by the size of input data, making it fundamentally unable to support extremely complex tasks like NLP which cannot fit in the shallow Hamiltonian. Therefore, rather than directly fitting data into a shallow DS model as NPGL does, we propose a novel mapping method that enables the deployment of existing deep neural networks (NNs) to DS machines. This allows us to leverage well-established, mature model architectures, rather than working solely with the DS model itself. To the best of our knowledge, this is the first work to map deep NNs—designed for traditional computing architectures—onto DS machines, with LLMs chosen as a specific application.
>
> ## 2. Key advantages of our approach (question 1-2):
> We deeply appreciate for the constructive suggestions. We have highlighted the key advantages of our approach over prior work and other emerging compute alternatives in the Appendix as a discussion section, which are also listed as below:
>
> Prior works (e.g. NPGL): Previous works, such as NPGL, focus on building new models based on the Hamiltonian. While this approach holds promise, it doesn’t directly benefit the computation of existing models like LLMs. In contrast, our approach maps existing models to DS machines, enabling us to transfer well-researched, mature models like LLMs to a powerful emerging computing framework. This allows for dramatic reductions in both training and inference costs, leveraging the power of DS machines without the need to reinvent models from scratch.
>
> ### Quantum computing:
> Quantum computing is a promising avenue but is still constrained by scalability issues and the need for complex error correction. As quantum systems scale, errors and noise increase, demanding advanced error-correction techniques that are not yet fully mature. Existing largest quantum computer has about 1100 qubits [1], which is too small to support LLM computing tasks. Meanwhile, due to the technology requirements, nowadays building and running quantum computers are still very expensive. In the contrast, DS machines are built on CMOS-compatible technologies, which are highly mature and the cost of its fabrication is at the same level of digital processors.
>
> ### Optical Computing:
> The optical computing solutions also rely on special technology and is hard to be integrated with digital systems. Building complex and accurate optical circuits can be a big challenge and very expensive, making the stability and feasibility of optical computer a larger challenge than DS machines.
>
> ### Computing-In-Memory (CIM):
> CIM technology is relatively feasible and CMOS compatible.  Existing CIM works only supports inference, achieving up to 100s times acceleration and 2000 times energy reduction [2], which is much lower than our solution especially on energy reduction.  The insight behind this is that CIM still follows a traditional instruction-based paradigm, completing computations step by step. In contrast, DS machines are driven by physical processes, specifically natural annealing, which doesn’t require step-by-step control. This allows DS machines to naturally perform complex tasks automatically without extra energy for controlling, achieving a much higher energy efficiency.
>
> ### Reference:
> [1] https://www.ibm.com/quantum/blog/quantum-roadmap-2033
>
> [2] Memory Is All You Need: An Overview of Compute-in-Memory Architectures for Accelerating Large Language Model Inference (https://arxiv.org/pdf/2406.08413 )
>
> ----------------------------
> * Due to limitation of characters, responses to other questions will be shown in next comments.

---

> > ### Author Response · Authors · 2024-11-21
> > **Part2 of Response to Reviewer HAWr**
> >
> > ## 3. Robustness on Circuit non-idealities (question 2-1)
> > Thank you very much for the insightful comments. We have incorporated a discussion on the robustness of DS machines on circuit non-idealities in the Appendix, along with new experimental results. We address two types of non-idealities: dynamic noise (e.g., thermal noise) and static offset (e.g., fabricated non-linearity and mismatch).
> >
> > ### Dynamic noise:
> > It has been proved that stochastic process like the natural annealing in DS machines is usually more robust to noise than deterministic process [1]. We further provide an experiment to evaluate the impact of varying noise levels on the DS machines. Dynamic noise was modeled as standard Gaussian noise with a standard deviation ranging from 0.05 to 0.15. The results demonstrate that DS machines exhibit high robustness to dynamic noise.
> >
> > | Model          | Noise level | MRPC   | QQP    | rte    | qnli   | sst2    |
> > |----------------|-------------|--------|--------|--------|--------|---------|
> > | gpt2-DS-ECL    | 0           | 76.91% | 89.24% | 63.11% | 87.94% | 90.82%  |
> > | gpt2-DS-ECL    | 5%          | 76.84% | 89.20% | 63.02% | 87.83% | 90.77%  |
> > | gpt2-DS-ECL    | 10%         | 76.55% | 88.93% | 62.71% | 87.56% | 90.33%  |
> > | gpt2-DS-ECL    | 15%         | 75.89% | 88.12% | 62.05% | 86.98% | 89.84%  |
> > | OPT1.3B-DS-ECL | 0           | 86.57% | 88.59% | 78.05% | 91.45% | 91.81%  |
> > | OPT1.3B-DS-ECL | 5%          | 86.43% | 88.52% | 77.99% | 91.40% | 91.75%  |
> >
> > ### Static offset:
> > Static offset is a common challenge in analog circuits, arising from factors like hardware non-linearity and mismatches during fabrication. Our proposed online training method addresses this effectively by training and performing inference on the same hardware device. This ensures that the model inherently adapts to the hardware’s biased patterns during training, resulting in accurate inference despite non-idealities.
> >
> > Moreover, when deploying a pre-trained model on different devices, a promising solution is to fine-tune the model on the target hardware for a few batches. This allows the model to quickly adapt to the specific biased pattern of the new hardware. To support this, we have added an experiment (shown in Fig. 7 in Appendix), which illustrates the recovery of model accuracy during the fine-tuning process, underscoring the practicality and effectiveness of this approach.
> >
> > ### Reference:
> > [1] Reasons for the superiority of stochastic estimators over deterministic ones: robustness, consistency and perceptual quality (https://dl.acm.org/doi/10.5555/3618408.3619511)
> >
> > ## 4. Tradeoff on Taylor expansion (question 2-2)
> > Thank you very much for your suggestions which help a lot for us to improve the paper. We have now included a thorough analysis and experimental evidence in the Appendix to explain our choice of a 3rd-order Taylor expansion.
> >
> > Before training the models, we conducted a pre-experiment by fine-tuning a pre-trained GPT-2 model on a small dataset (Shakespeare, 300k) with different orders of Taylor expansion. The results show that the 3rd-order expansion achieves sufficiently low validation loss, while the improvement from higher orders is marginal. In terms of hardware requirements, the resource cost scales approximately linearly with the order of each term.  For instance, a polynomial like “x^3+x^2+x” requires around 6 resource units, while adding a term like “x^5” would demand an additional 5 units. Based on this trade-off, we selected the 3rd-order expansion as the most balanced design choice. For those prioritizing accuracy over hardware efficiency, higher-order expansions can be adopted and are compatible with our framework.
> >
> > | Tayler Expansion order     | Validation loss | Loss Drop | Hardware costs (units)  |
> > |----------------------------|-----------------|-----------|-------------------------|
> > | Inf (baseline exponential) | 3.25            | 0         | -                       |
> > | 1                          | 3.35            | 0.10      | 1                       |
> > | 3                          | 3.28            | 0.03      | 6                       |
> > | 5                          | 3.27            | 0.02      | 15                      |
> > | 7                          | 3.27            | 0.02      | 28                      |
> >
> > -------
> > * Due to limitation of characters, responses to other questions will be shown in next comments.

---

> > > ### Author Response · Authors · 2024-11-21
> > > **Part3 of Response to Reviewer HAWr**
> > >
> > > ## 5. Model scalability problem (question 3).
> > > Thank you for this insightful question. We have added a discussion and analysis in the revised manuscript to address this point. To ensure simulating the DS machines behavior correctly, we use a very accurate simulation which is very time-consuming, and it’s out of our computing capability to evaluate a 405B model.  We have tried our best to include two larger models up to 7B in the evaluation section to demonstrate that our solution exhibits good scalability.  The new evaluation results are added in Table 1, Table 2, and Table 3.
> > >
> > > As a theoretically analysis, DS machines have been shown to achieve linear complexity with respect to the number of nodes [1], making them inherently scalable with increased chip area. For reference, the backbone DS machine [2] only occupies about 5 mm² of area, whereas modern GPUs like the H100 have a die size of 814 mm², and wafer-scale chips—such as those with up to 46,255 mm²—are emerging [3]. Based on linear complexity, a single-chip solution could theoretically support millions of nodes within a single DS machine. Second, for even larger models or faster training, multi-chip approaches offer a viable path forward. Existing research has explored multi-chip solutions for DS machines [4], where individual chips perform annealing with periodic synchronization. We are also investigating promising techniques such as deploying models across multiple DS machine chips to achieve pipeline parallelism.
> > >
> > > ### Reference:
> > > [1] DS-GL: Advancing Graph Learning via Harnessing Nature’s Power within Scalable Dynamical Systems
> > >
> > > [2] Extending Power of Nature from Binary to Real-Valued Graph Learning in Real World
> > >
> > > [3] Wafer-Scale Computing: Advancements, Challenges, and Future Perspectives [Feature]
> > >
> > > [4] Increasing ising machine capacity with multi-chip architectures
> > >
> > > ## 6. Details of setup (question 4-1)
> > > Thank you for your insightful questions. The reported 1,000× speedup represents the average improvement across all downstream datasets. For each dataset, we compared the total training or inference time required to process the entire training or evaluation set. Energy consumption was similarly evaluated for the entire dataset. The core reason behind this significant improvement is that DS machines inherently perform an annealing process driven by analog currents, bypassing the need for step-by-step digital instructions. This natural, continuous computation enables DS machines to achieve exceptional speed and energy efficiency.
> > >
> > > ## 7. Emerging GPU optimizations(question4-2)
> > > In the previous comparison, we did not incorporate additional GPU optimization technologies. We have clarified in the revised paper that the comparison was based on a naïve GPU implementation and introduced emerging GPU optimization works like vLLM in the paper. Meanwhile we also added more details in the setup section specifying that the batch size for inference matches that used in training. The reason we select a naive implementation is that there are lots of GPU optimization works, but digital-based works theoretically cannot achieve a too high speedup – even with 100% utility of GPU, the speedup is still less than 10 times, whereas DS machines deliver orders-of-magnitude improvements. In other words, this work is not on the same track with the GPU optimization works which are more practical and can directly use on existing hardware infrastructures. The target of this work is to provide a new solution for future exploration, aiming to fundamentally solve the increasing computing demanding of LLMs.
> > >
> > > ## 8. Meaningful Inference Metrics (question4-3)
> > > Thank you very much for introducing these meaningful metrics to us and we have updated them in the revised manuscript for better evaluation.
> > >
> > > Table7: Additional Inference Metrics on average of all datasets
> > > |  | Time to First Token (s) | Token Generation Rate (token/s) | Energy Efficiency (token/KWh)  |
> > > |---|---|---|---|
> > > | GPT2 | 1.31E-4 | 7.62E+3 | 6.84E+7  |
> > > | GPT2-DS | 1.20E-6 | 8.33E+5 | 1.11E+12  |
> > > | gpt2-m | 3.92E-4 | 2.55E+3 | 2.38E+7  |
> > > | gpt2-m-DS | 2.50E-6 | 4.00E+5 | 5.33E+11  |
> > > | OPT1.3B | 6.85E-4 | 1.46E+3 | 1.37E+7  |
> > > | OPT1.3B-DS | 6.00E-6 | 1.67E+5 | 2.22E+11  |
> > > | OPT2.7B | 2.12E-3 | 4.72E+2 | 4.34E+6  |
> > > | OPT2.7B-DS | 1.30E-5 | 7.69E+4 | 1.03E+11  |
> > > | Llama2-7B | 7.08E-3 | 1.41E+2 | 1.35E+6  |
> > > | Llama2-7B-DS | 3.30E-5 | 3.03E+4 | 4.04E+10  |
> > >
> > > ## 9. Comparison with prior works and other emerging technologies (question4-4)
> > > Thank you for your comments, please refer to response 2 where we discuss a detail comparison with prior works and other emerging technologies.
> > > ## 10. Revision of Abstract (recommendation)
> > > Thank you for your reasonable suggestion and we have abbreviated the abstract to briefly explain the paper’s novelty and contributions.

---

> ### Comment · Reviewer_HAWr · 2024-11-21
>
> Thank you for your responses.
>
> I still feel my major concerns are not yet clearly addressed in the updated revision of the papers (main sections not just Appendix). I am not a fan of how the authors *boast* of such significant speedup and energy efficiency comparing between a mature technology (GPUs) with an emerging one (Dynamical Systems) that still is immature, not fully grounded, and possibly impractical. My concerns are further exacerbated by the fact that when deriving these comparision numbers, the authors have not fully leveraged existing SOTA techniques (such as vLLM [1] which easily improves throughput 2-4x), leading to possibly severly under-reported numbers for GPUs. Specifically,
>
> 1. I share similar concerns with Reviewer xU44 on *examples of real-world applications* and particularly **physical practicality** of the proposed solution. While in the authors did try to clarify, [2,3] only seemed to present theoretical analysis based on simluations without physical implementations, and [4] showed physical implementations solving MAX-CUT on small graphs and not so impressive (or even advantageous) runtimes. This leads me to question the feasiblity of such solutions and whether such systems would have the performance if physically implemented versus what the authors claim to be theoretically possible.
>
> 2. Concerns on unfair comparisions. The authors have not fully leveraged existing SOTA techniques (such as vLLM [1] ), leading to possibly severly under-reported numbers for GPUs. Furthermore, the authors are comparing between **a measured performance** of GPUs to that of **theoretical numbers on paper** of an emerging tech. I do not think such comparisions are solid, and I also do not like how they are presented in the paper. For example authors claim 37,545x speedup on inference, this number could possibly easily been cut to more than 1/2 with techniques such as vLLM, and more if accounting for what is theoretically possible versus physically implementable. I do not think it should be a game of what tech presents the most amazing theoretical improvement, and in fact I would appreciate it if the authors responsibly *scale back* on such claims with full disclosure on the technology's limitations (and that some of the numbers presented might be based on unfair comparison of theoretical v.s. physical).
>
> 3. Concerns on scalability. I do appreciate the authors efforts on adding experiments for larger 7B models and understand the limitations of not being able to scale experiments to even larger models. I would think it to be much better even if the authors in the main section of the paper to analyze the cost and performance of scaling on such systems theoretically. Possibly it would also help with visualizations if some key metrics of Table 2 and Table 3 are presented in a plot (with theoretical projections on how such scale to larger models). It would also be helpful if a scaling plot on the cost of scaling (area etc.) or explained in text.
>
> 4. Meaningful metrics. I appreciate the authors updates on tokens/s and tokens/kWh. I suggest replacing Table 2, Table 3 (or suggested plots above) with these metrics.
>
> 5. No disclosure in the main paper on limitations. Given my concerns above (see 1,2,3), I suggest the authors fully disclose the limitations of such emerging technology responsibly throughout their main paper (including abstract, intro, results, and if possible including an additional section on limitations). Especially I have questions on, (1) are such systems even physically possible to reach what the authors are proposing? (2) if they could, would they even perform to what is theoretically possible as stated?; (3) are there challenges in scaling, and if so are there solutions?; (4) how difficult would it be to implement such a system at scale to what is currently achieved with GPUs? What is the current progress on physically implementing such systems and how do they compare to the scale of GPUs?
>
> With the above stated, I am open to further discussions and open to increasing my ratings if the authors clarifications (on my concerns of limitations, comparisions, and scalability) do get reflected in the main sections of the paper. As of the current state of the paper and authors responses to my concerns, I could not warrant a higher rating.
>
> *References*
>
> [1] Efficient Memory Management for Large Language Model Serving with PagedAttention
>
> [2] Ising-Traffic: Using Ising Machine Learning to Predict Traffic Congestion under Uncertainty.
>
> [3] Extending Power of Nature from Binary to Real-Valued Graph Learning in Real World.
>
> [4] Experimental investigation of performance differences between coherent Ising machines and a quantum annealer.

---

> > ### Author Response · Authors · 2024-11-21
> >
> > Thank you very much for your detailed feedback and thoughtful suggestions. We completely agree that for early-stage research on emerging technologies, it is essential to clearly outline the limitations and challenges to help readers assess the reliability and practicality of the approach. We are committed to addressing your concerns and revising the paper to better articulate these points, ensuring it serves as a strong foundation for future exploration.
> >
> > We aim to submit the updated revision within two days and sincerely appreciate your valuable input in helping improve this work. Thank you again for your efforts!

---

> > > ### Comment · Reviewer_HAWr · 2024-11-21
> > >
> > > I look forward to your revisions. Please comment once ready so I can be notified of your update.

---

> > > > ### Author Response · Authors · 2024-11-24
> > > > **Part1 of Response to Reviewer HAWr**
> > > >
> > > > ## Overview
> > > > We sincerely appreciate you againfor your insightful comments. We have addressed your concerns in the main paper of the new updated revision and highlight the modifications in blue (the previous revision is marked in green).  Before listing the specific response to your questions, we want to first share some thoughts with the reviewers.
> > > >
> > > > The DS Machine represents a groundbreaking new type of computing architecture that has emerged over the past five years. Its exceptional performance has been demonstrated in solving NP-complete and NP-hard optimization problems, and the research community is now exploring its potential in artificial intelligence (AI), particularly leveraging its intrinsic dynamical system properties for efficient sampling. The CMOS capacitor-based DS Machine, introduced in 2021, has garnered significant attention due to its manufacturing simplicity, compact chip area, and low power consumption.
> > > >
> > > > Although capacitor-based DS Machines have been successfully fabricated and tested, they currently operate at a relatively small scale, with implementations supporting thousands of spins/nodes (equivalent to million-level parameters). Nonetheless, we recognize its potential to overcome existing bottlenecks in AI computation, particularly in training, and are committed to demonstrating this potential through advanced modeling and simulation. This foundational work will facilitate securing the resources necessary for developing large-scale physical prototypes capable of operating at the scale required for large language models (LLMs).
> > > >
> > > > Realizing the efficiency reported in theoretical studies with a GPU-scale chip has the potential to mark a transformative milestone in the future, paving the way for a new era of AI computation.
> > > >
> > > > ## 1. Physical practicality Concern
> > > > Thank you very much for your insightful comments. We have added more detailed discussion about the physical practicality of our solution in a new discussion section in the main paper. The response is outlined below:
> > > >
> > > > CMOS-based DS machines have various implementations, including digital-based [1], latch-based [2], and oscillator-based [3] designs, all validated as viable computing approaches through physical prototypes. In this work, we adopt a CMOS-capacitor-based DS machine as the backbone—a newly emerging DS machine design that has been physically manufactured and tested at a small scale with thousands of spins (millions of parameters). While large-scale physical prototype is still under manufacture/test, we are confident in its feasibility for the established components and technologies replied on in the design.
> > > >
> > > > Specifically, the CMOS-capacitor-based DS machine shares its core components and technologies with other CMOS-based DS machines. The primary difference lies in using capacitors to represent spins instead of latches or oscillators. Additionally, CMOS capacitors have been widely used in analog computing technologies, such as [4], reinforcing the practicality of capacitor-based spins.
> > > >
> > > > Beyond the feasibility of this specific DS machine based on CMOS capacitors, we would like to emphasize that the proposed mapping solution that apply LLM to DS machines is a general solution that can be applied to various DS machine variants. Specifically, the proposed mapping method transforms LLM computation as Hamiltonian-based natural annealing which is a foundation of all variants of CMOS-based DS-Machine, enabling the use of DS machines in general in LLM acceleration.
> > > >
> > > > ### Reference
> > > > [1] A Million Spiking-Neuron Integrated Circuit with a Scalable Communication Network and Interface
> > > >
> > > > [2] CTLE-Ising: A Continuous-Time Latch-Based Ising Machine Featuring One-Shot Fully Parallel Spin Updates and Equalization of Spin States
> > > >
> > > > [3] A 1,968-node coupled ring oscillator circuit for combinatorial optimization problem solving
> > > >
> > > > [4] A 2.5GHz 7.7TOPS/W Switched-Capacitor Matrix Multiplier with Co-designed Local Memory in 40nm
> > > >
> > > > -------
> > > > Due to limitation of characters, responses to other questions will be shown in next comments.

---

> > > > > ### Author Response · Authors · 2024-11-24
> > > > > **Part2 of Response to Reviewer HAWr**
> > > > >
> > > > > ## 2. Concerns Regarding Fair Comparisons and Scalability
> > > > > We sincerely thank you for your valuable feedback, which has helped us significantly improve the evaluation section of the manuscript. Below, we outline our responses and the improvement made in the revised version:
> > > > >
> > > > > ## Fair Comparisons:
> > > > > We have incorporated vLLM to enhance GPU inference performance, providing a fairer comparison. Additionally, we explicitly clarified in the main paper that the evaluation of our solution is through simulation and not based on physical measurements. The updated comparison results are based on the metrics per your insightful suggestion.
> > > > >
> > > > > Table 2: Performance comparison between GPU and DS machines.
> > > > > |  | Training | Training | Inference | Inference | Inference |
> > > > > |---|:---:|:---:|:---:|:---:|:---:|
> > > > > | Metric | Training Throughput (token/s) | Energy Efficiency (token/KWh) | Time to First Token (s) | Token Generation Rate (token/s) | Energy Efficiency (token/KWh) |
> > > > > | GPT2 | 1.37E+04 | 6.19E+07 | 6.46E-05 | 1.55E+04 | 1.39E+08 |
> > > > > | GPT2-DS | 6.70E+06 | 8.93E+12 | 1.20E-06 | 8.33E+05 | 1.11E+12 |
> > > > > | gpt2-m | 3.24E+03 | 1.46E+07 | 1.80E-04 | 5.55E+03 | 5.18E+07 |
> > > > > | gpt2-m-DS | 3.27E+06 | 4.36E+12 | 2.48E-06 | 4.03E+05 | 5.38E+11 |
> > > > > | OPT1.3B | 1.20E+03 | 5.41E+06 | 2.76E-04 | 3.62E+03 | 3.41E+07 |
> > > > > | OPT1.3B-DS | 3.08E+06 | 9.56E+11 | 2.55E-06 | 3.92E+05 | 1.22E+11 |
> > > > > | OPT2.7B | 3.97E+02 | 1.78E+06 | 8.12E-04 | 1.23E+03 | 1.13E+07 |
> > > > > | OPT2.7B-DS | 2.33E+06 | 4.37E+11 | 3.41E-06 | 2.93E+05 | 5.50E+10 |
> > > > > | Llama2-7B | 1.34E+02 | 6.05E+05 | 2.51E-03 | 3.98E+02 | 3.82E+06 |
> > > > > | Llama2-7B-DS | 2.24E+06 | 1.56E+11 | 3.57E-06 | 2.80E+05 | 1.95E+10 |
> > > > >
> > > > >
> > > > > ## Scalability Analysis:
> > > > > To address concerns about scalability, we conducted a detailed evaluation of two potential scaling strategies for DS machines: single-chip (scale up) and multi-chip (scale out) solutions.
> > > > >
> > > > > ### Single-Chip Solution (Scale up):
> > > > > The chip area and power consumption increase with model size. We evaluate the token generation rate, energy efficiency, and hardware cost for models up to 66B parameters (models larger than 7B are estimated). In the main paper, we acknowledged the practical challenges of scaling solely through single-chip solutions for extremely large models due to physical constraints such as process yield and thermal management. For instance, the estimated chip area for a 66B model surpasses that of a typical GPU chip. Though the industry has built some wafer-scale extremely large chips [5], multi-chip scaling out solution is a more mature and widely used solution in AI accelerator industry.  Therefore, we recommend and evaluate multi-chip scaling out solution as a more feasible alternatives for extreme large models.
> > > > >
> > > > > ### Multi-Chip Solution (Scale out):
> > > > > We have highlighted in the main paper that while multi-chip solution involves fixed chip sizes, additional communication overhead will inevitably reduce the system efficiency. We evaluated performance for models ranging from 1.3B to 405B parameters (estimates for models larger than 7B). Results indicate that communication costs marginally degrade performance.
> > > > >
> > > > > We visualized the scaling trends for both solutions (in Fig. 7), illustrating near-linear reductions in token generation rate and energy efficiency as model size increases. The multi-chip solution demonstrated more practical performance for larger models. The revised manuscript includes a figure highlighting these trends and an additional table summarizing the scaling of chip area and power consumption (in Table. 3). We also highlight that the performance of actual multi-chip systems can be influenced by the implementation of memory hierarchy, communication bandwidth, and many system level trade-offs.
> > > > >
> > > > > Table 3: Hardware cost with single-chip scaling.
> > > > > | Model size | Area (mm2) | Power consumption (w) |
> > > > > |---|---|---|
> > > > > | GPT2-DS | 19.8 | 2.7 |
> > > > > | gpt2-m-DS | 19.8 | 2.7 |
> > > > > | OPT1.3B-DS | 83.2 | 11.6 |
> > > > > | OPT2.7B-DS | 131.1 | 19.2 |
> > > > > | Llama2-7B-DS | 332.4 | 51.6 |
> > > > > | OPT30B-DS(estimated) | 907.5 | 138.2 |
> > > > > | OPT66B-DS(estimated) | 1448.6 | 226.7 |
> > > > >
> > > > > We believe these updates address your concerns and significantly enhance the clarity and rigor of the scalability discussion. Thank you again for your thoughtful feedback, which has been invaluable in improving this work.
> > > > >
> > > > > ### Reference:
> > > > > [5] Wafer-Scale Computing: Advancements, Challenges, and Future Perspectives [Feature] (https://ieeexplore.ieee.org/abstract/document/10460211)
> > > > >
> > > > > ## 3. Meaningful metrics
> > > > > Thank you again for your invaluable suggestions, we have replaced the evaluation tables with the new metrics per the reviewer’s suggestions.
> > > > >
> > > > > ----------
> > > > > Due to limitation of characters, responses to other questions will be shown in next comments.

---

> > > > > > ### Author Response · Authors · 2024-11-24
> > > > > > **Part3 of Response to Reviewer HAWr**
> > > > > >
> > > > > > ## 4. Limitation analysis in main paper.
> > > > > > We have revised the main paper to incorporate analysis on the limitations of this emerging technology, including abstract, intro, evaluation results, and an additional discussion section. We have marked all the modifications in blue in the revision, and we outlined the response to your specific questions as below:
> > > > > >
> > > > > > ### (1) are such systems even physically possible to reach what the authors are proposing?
> > > > > > As addressed in Response 1, the backbone capacitor-based DS machine has been manufactured and tested at a small scale. For large-scale manufacture, the feasibility is supported by the established components and technologies it employs. Furthermore, our mapping solution is hardware-agnostic, offering high adaptability to other DS machine variants.
> > > > > >
> > > > > > ### (2) if they could, would they even perform to what is theoretically possible as stated?
> > > > > > The performance of other DS machine variants has been verified in numerous studies, demonstrating their practical capabilities. Similarly, the backbone DS machine used in this work has shown promising results at small scales in AI and optimization problems. It is worth emphasizing that the backbone DS machine is built on mature and widely used analog technologies, with commercial circuit simulation tools like Cadence Virtuoso serving as reliable design platforms. While minor differences between simulated and actual chip measurements are inevitable, they are typically minimal, especially when using proven technologies.
> > > > > >
> > > > > > ### (3) are there challenges in scaling, and if so are there solutions?
> > > > > > As discussed in Response 2, we acknowledge in the evaluation section that scaling a single chip to extremely large sizes, such as beyond a GPU’s die size, poses practical challenges. To address this, we recommend multi-chip solutions (e.g. through chiplet), where each chip maintains a manageable size, a common practice in modern computing systems especially for commercial AI accelerators. While multi-chip systems introduce communication overhead, leading to marginal performance reduction, we have evaluated and included this impact in our analysis. This ensures a realistic assessment of scalability while showcasing the viability of multi-chip architectures.
> > > > > >
> > > > > > ### (4) how difficult would it be to implement such a system at scale to what is currently achieved with GPUs? What is the current progress on physically implementing such systems and how do they compare to the scale of GPUs?
> > > > > >
> > > > > > We have added this limitation and potential solution in the discussion section in main paper, and the response are outlined as below:
> > > > > >
> > > > > > The study of DS machines is still in its early stages, focusing primarily on their individual architecture, algorithms, and computation methods. To fully realize their potential, it will be crucial to integrate DS machines into existing AI infrastructures. Notably, DS machines are fully CMOS-compatible and operate at room temperature, enabling their integration as co-processors within the established protocols and workflows used for FPGAs, TPUs, and other commercial AI accelerators. This standardized flow typically comprises three primary components: **Front End**, **Dispatcher**, and **Backend**. The Front End provides execution APIs designed specifically for DS machines, offering advanced functionalities such as natural annealing, continuous training, and efficient data and parameter loading. The Dispatcher acts as an intermediary, mapping these APIs to corresponding backend operations, abstracting hardware complexities, and simplifying user interaction with DS machine functionalities. The Backend typically includes a robust runtime and four key components: Data Compilation, to optimize and prepare inputs for computation; Data Structures, to design efficient formats for storing and managing data; Operators, defining core computation functionalities; and Device Drivers, facilitating direct communication with DS machine hardware.

---

> ### Comment · Reviewer_HAWr · 2024-11-25
> **Updates on Scores**
>
> Thanks for the authors responses.
>
> I have raised my scores significantly from 3 to 6. The authors have greatly revised their draft and provided relevant information in the discussion. Specifically improvements were made:
> - Discussions on Limitations: The authors have now avoided reporting comparision numbers directly of the theoretical results on DS Machines with measured performance of GPUs. The authors have modified the main paper (abstract, introduction) to reflect such changes. The authors also provide a new Section 5 (and Appendix A1) to discuss the limitation concerns of DS machines.
> - Improved baselines and metrics: The authors have updated to use vLLM as GPU baselines for throughput and energy. The authors have also updated Table 2 to report on more relevant metrics such as token/s and token/KWh
> - Concerns on scalability and cost: The authors have provided a better visualization (Figure 7) with theoretical projections on scalaing curve on training/inference performance. The authors also discuss cost of scaling in Table 3.
> - Concerns on nonidealities: The authors provide such studies in Appendix regarding trade-offs on Taylor expansion and robustness on circuit non-idealities such as dynamic noise and static offset.
>
> Reasons on not a higher score. Although I find the work to be interesting and quite possibly the first work to customize DS machines to LLMs, I fail to find too much of a distinction in novelty compared with prior work. While the authors state their mapping method to be the main contribution (Section 3.2), I do not find it as a considerable ''novelty'' factor. I do not consider myself an expert in this domain (especially DS machines), thus I have low confidence and also reluctant to offering a higher score.
>
> Based on all the above, updated version of the paper, the good efforts authors took to address my concerns, I have raised my score to 6.

---

> ### Author Response · Authors · 2024-11-25
>
> We are thrilled to hear that our revisions have addressed your concerns. We deeply value your insightful and constructive feedback and sincerely appreciate your recognition and appreciation of our work, which affirm our efforts in exploring this emerging area. Once again, we thank you for your thoughtful inputs and for acknowledging the potential impact of our contributions!

---

> > ### Author Response · Authors · 2024-11-26
> >
> > The discussions over the past few days have been incredibly insightful and have greatly contributed to improving our paper. Much appreciated for the reviewer's suggestions. With the extension of the discussion period, we see this as a rare and invaluable opportunity to further engage with the reviewer and learn from their expertise and wisdom.
> >
> > Therefore, we would like to take this opportunity to clarify the new contributions of this work compared to prior art (e.g., NP-GL), beyond the mapping of LLM kernels onto DS machines. Specifically, DS-LLM (this work) represents the first effort to enable on-device training on a dynamical systems (DS) machine, thereby extending the computational capabilities of DS machines from inference to training. In contrast, prior works restricted DS machines to inference tasks, requiring training to be conducted on GPUs.
> >
> > Once again, we deeply appreciate the reviewer’s recognition of the potential of our work in advancing LLM development and their constructive feedback throughout this process. We are always eager to receive any additional suggestions.

---

### Official Review · Reviewer_v14i · 2024-10-31

**Soundness:** 3
**Presentation:** 3
**Contribution:** 3
**Rating:** 6
**Confidence:** 2

**Summary:**

This paper proposes a novel approach for constructing large language models (LLMs) using analog dynamic system (DS) machines, offering an alternative to conventional CMOS-based hardware. The authors present two methods for mapping existing Transformer-based models onto DS machines, thoroughly analyzing key components within Transformers to facilitate a complete transformation compatible with DS architecture. The proposed DS-based models achieve significantly faster training and inference times, along with notably reduced energy consumption compared to traditional CMOS-based systems.

**Strengths:**

* The paper introduces a creative application of analog DS machines for LLM construction, advancing research on energy-efficient AI model deployment.
* By analyzing and adapting key Transformer components for DS architecture, the authors provide a clear methodology for how LLMs can be mapped effectively, which contributes to the feasibility of DS-based machine learning.
* The substantial reductions in training and inference time and energy consumption demonstrate the practical advantages of DS machines over CMOS for LLMs.

**Weaknesses:**

* The paper could benefit from a broader benchmarking analysis that includes different LLM sizes.
* While DS machines are promising in terms of efficiency, their limitations (e.g., stability and precision constraints) are not fully discussed.

**Questions:**

* As LLMs scale up, are additional considerations or modifications necessary for DS machines to handle larger model architectures effectively?

---

> ### Author Response · Authors · 2024-11-21
> **Part1 of Response to Reviewer v14i**
>
> ## Overview
>
> Thank you very much for your meaningful feedback which helps us a lot to improve the paper. We have updated a revision version of paper to address your concerns and you can find the contents labled as Reviewer C like "C-Q1".  The detailed response to your comments are listed below.
>
> ## 1. Limitations of DS machines (weakness 2 and question):
>
> We deeply appreciate your insightful comments and have added a discussion section in the Appendix to address the limitations and challenges associated with DS machines. Below, we outline our responses to the three main concerns raised in your feedback: stability, precision constraints, and scalability.
>
> ### Stability:
>
> The stability of DS machines is inherently promising due to their fundamental characteristics. Compared to modern GPUs, DS machines consume significantly less power, which reduces the risk of overheating. Additionally, as DS machines rely on a stochastic annealing process rather than deterministic computation, they are naturally more robust to noise and non-ideal factors [1]. To further support this, we included a new experiment to evaluate the impact of noise on the accuracy of DS machines. The results demonstrate that DS machines exhibit high stability and robustness under various noise conditions.
>
> | Model          | Noise level | MRPC   | QQP    | rte    | qnli   | sst2    |
> |:--------------:|:-----------:|:------:|:------:|:------:|:------:|:-------:|
> | gpt2-DS-ECL    | 0           | 76.91% | 89.24% | 63.11% | 87.94% | 90.82%  |
> |  gpt2-DS-ECL              | 5%          | 76.84% | 89.20% | 63.02% | 87.83% | 90.77%  |
> |  gpt2-DS-ECL              | 10%         | 76.55% | 88.93% | 62.71% | 87.56% | 90.33%  |
> |  gpt2-DS-ECL              | 15%         | 75.89% | 88.12% | 62.05% | 86.98% | 89.84%  |
> | OPT1.3B-DS-ECL | 0           | 86.57% | 88.59% | 78.05% | 91.45% | 91.81%  |
> | OPT1.3B-DS-ECL               | 5%          | 86.43% | 88.52% | 77.99% | 91.40% | 91.75%  |
> |  OPT1.3B-DS-ECL              | 10%         | 86.15% | 88.31% | 77.63% | 91.22% | 91.43%  |
> |  OPT1.3B-DS-ECL              | 15%         | 85.67% | 87.75% | 77.24% | 90.89% | 91.01%  |
>
> ### Precision Constraints:
>
> The precision of computation in DS machines is inherently continuous due to their analog nature. However, the precision of Analog-to-Digital Converters (ADCs) and Digital-to-Analog Converters (DACs) used in the system do affect the overall accuracy. Fortunately, ADCs and DACs are well-established technologies with numerous mature solutions that allow for various design trade-offs—for instance, achieving 16-bit precision with lower power consumption or 32-bit precision with higher power consumption. Such trade-offs align with and are theoretically compatible with existing quantization techniques, providing flexibility to balance precision and power efficiency based on the application requirements.
>
> ### Scalability Issues:
>
> While prior work like NP-GL was designed for small graphs with fewer than 1,000 nodes, the capacity of DS machines can be significantly expanded to handle much larger scales. First, DS machines have demonstrated linear complexity with respect to the number of nodes [2], making them inherently scalable with increased chip area. For context, NP-GL occupies only about 5 mm², whereas modern GPUs like the H100 have a die size of 814 mm², and wafer-scale chips—such as those with up to 46,255 mm²—are emerging [3]. Based on linear complexity, a single-chip solution could theoretically support millions of nodes within a single DS machine.
>
> Second, for even larger models or faster training, multi-chip approaches offer a viable path forward. Existing research has explored multi-chip solutions for DS machines [4], where individual chips perform annealing with periodic synchronization. We are also investigating promising techniques such as deploying models across multiple DS machine chips to achieve pipeline parallelism.
>
> Overall, the scalability of DS machines is theoretically well-founded, offering both single-chip and multi-chip solutions to meet the demands of increasingly large and complex models.
>
> ### Reference:
> [1] Reasons for the superiority of stochastic estimators over deterministic ones: robustness, consistency and perceptual quality
>
> [2] DS-GL: Advancing Graph Learning via Harnessing Nature’s Power within Scalable Dynamical Systems
>
> [3] Wafer-Scale Computing: Advancements, Challenges, and Future Perspectives [Feature]
>
> [4] Increasing ising machine capacity with multi-chip architectures
>
> ## 2. Comparison under more LLM sizes (weakness 1)
> We sincerely appreciate your insightful suggestions. In response, we have included evaluations on two larger-sized models including OPT-2.7B and Llama2-7B to better assess our solution's performance. The results demonstrate that DS-LLM continues to offer a comparable accuracy and significant speedup & energy reduction over traditional GPU-based solutions.
>
> The updated tables are shown in next  comments.

---

> > ### Author Response · Authors · 2024-11-21
> > **Part2 of Response to Reviewer v14i**
> >
> > Table 1: Accuracy comparison (in Accuracy (%)): the higher the better
> > | Dataset            | MRPC       | QQP        | rte       | qnli      | sst2             |
> > |--------------------|------------|------------|-----------|-----------|------------------|
> > | Task               | paraphrase | paraphrase | Inference | Inference | Single-Sentence  |
> > | gpt2               | 75.00%     | 88.43%     | 63.18%    | 88.03%    | 91.63%           |
> > | gpt2-DS            | 76.72%     | 89.04%     | 63.54%    | 88.28%    | 91.29%           |
> > | gpt2-DS-ECL        | 76.91%     | 89.24%     | 63.11%    | 87.94%    | 90.82%           |
> > | gpt2-medium        | 79.66%     | 90.57%     | 68.59%    | 91.05%    | 93.58%           |
> > | gpt2-medium-DS     | 79.17%     | 90.55%     | 70.40%    | 90.33%    | 93.35%           |
> > | gpt2-medium-DS-ECL | 79.31%     | 90.09%     | 70.41%    | 89.73%    | 93.06%           |
> > | OPT1.3B            | 84.07%     | 90.94%     | 77.26%    | 91.09%    | 92.43%           |
> > | OPT1.3B-DS         | 87.01%     | 88.48%     | 78.70%    | 91.41%    | 92.20%           |
> > | OPT1.3B-DS-ECL     | 86.57%     | 88.59%     | 78.05%    | 91.45%    | 91.81%           |
> > | OPT2.7B            | 86.52%     | 91.03%     | 82.33%    | 93.15%    | 94.08%           |
> > | OPT2.7B-DS         | 86.54%     | 90.98%     | 82.48%    | 93.27%    | 94.04%           |
> > | OPT2.7B-DS-ECL     | 86.74%     | 90.67%     | 82.44%    | 93.41%    | 94.25%           |
> > | Llama2-7B          | 90.01%     | 91.10%     | 88.45%    | 95.75%    | 96.58%           |
> > | Llama2-7B-DS       | 89.47%     | 90.95%     | 88.70%    | 95.69%    | 96.32%           |
> > | Llama2-7B-DS-ECL   | 89.56%     | 91.08%     | 88.57%    | 95.73%    | 95.79%           |
> >
> > Table 2: Performance comparison on training time and energy consumption
> > | Metric           | training time |         |         |         |         | energy consumption |         |         |         |          |
> > |------------------|---------------|---------|---------|---------|---------|--------------------|---------|---------|---------|----------|
> > | Dataset          | MRPC          | QQP     | rte     | qnli    | sst2    | MRPC               | QQP     | rte     | qnli    | sst2     |
> > | GPT2             | 5.17E+1       | 2.12E+3 | 4.56E+1 | 8.32E+2 | 3.07E+2 | 4.14E+4            | 1.69E+6 | 3.65E+4 | 6.66E+5 | 2.46E+5  |
> > | GPT2-DS-ECL      | 4.44E-2       | 4.37E+0 | 3.00E-2 | 1.26E+0 | 8.04E-1 | 3.77E-1            | 3.71E+1 | 2.55E-1 | 1.07E+1 | 6.83E+0  |
> > | gpt2-m           | 1.62E+2       | 9.00E+3 | 1.55E+2 | 3.42E+3 | 1.32E+3 | 1.30E+5            | 7.20E+6 | 1.24E+5 | 2.74E+6 | 1.05E+6  |
> > | gpt2-m-DS-ECL    | 9.25E-2       | 9.10E+0 | 6.25E-2 | 2.63E+0 | 1.68E+0 | 7.86E-1            | 7.74E+1 | 5.31E-1 | 2.23E+1 | 1.42E+1  |
> > | OPT1.3B          | 1.78E+2       | 2.42E+4 | 1.77E+2 | 6.46E+3 | 2.83E+3 | 1.42E+5            | 1.94E+7 | 1.42E+5 | 5.17E+6 | 2.27E+6  |
> > | OPT1.3B-DS-ECL   | 2.22E-1       | 2.18E+1 | 1.50E-1 | 6.30E+0 | 4.02E+0 | 1.89E+0            | 1.86E+2 | 1.28E+0 | 5.36E+1 | 3.42E+1  |
> > | OPT2.7B          | 3.90E+2       | 7.34E+4 | 3.71E+2 | 2.02E+4 | 1.20E+4 | 3.12E+5            | 5.87E+7 | 2.97E+5 | 1.61E+7 | 9.59E+6  |
> > | OPT2.7B-DS-ECL   | 4.81E-1       | 4.73E+1 | 3.25E-1 | 1.37E+1 | 8.71E+0 | 4.09E+0            | 4.02E+2 | 2.76E+0 | 1.16E+2 | 7.40E+1  |
> > | Llama2-7B        | 1.55E+3       | 2.17E+5 | 1.49E+3 | 6.27E+4 | 3.46E+4 | 1.24E+6            | 1.73E+8 | 1.19E+6 | 5.02E+7 | 2.77E+7  |
> > | Llama2-7B-DS-ECL | 1.22E+0       | 1.20E+2 | 8.25E-1 | 3.47E+1 | 2.21E+1 | 1.04E+1            | 1.02E+3 | 7.01E+0 | 2.95E+2 | 1.88E+2  |
> >
> > *Another table will be shown in next comment

---

> > > ### Author Response · Authors · 2024-11-21
> > > **Part3 of Response to Reviewer v14i**
> > >
> > > Table 3: Performance comparison on inference time and energy consumption.
> > >
> > > | Metric       | inference time |         |         |         |         | energy consumption |         |         |         |          |
> > > |--------------|----------------|---------|---------|---------|---------|--------------------|---------|---------|---------|----------|
> > > | Dataset      | MRPC           | QQP     | rte     | qnli    | sst2    | MRPC               | QQP     | rte     | qnli    | sst2     |
> > > | GPT2         | 1.38E-1        | 1.17E+1 | 1.43E-1 | 1.84E+0 | 2.84E-1 | 1.10E+2            | 9.37E+3 | 1.14E+2 | 1.47E+3 | 2.27E+2  |
> > > | GPT2-DS      | 2.04E-3        | 4.69E-1 | 3.60E-3 | 6.48E-3 | 2.16E-3 | 5.51E-3            | 1.27E+0 | 9.72E-3 | 1.75E-2 | 5.83E-3  |
> > > | gpt2-m       | 4.46E-1        | 3.79E+1 | 3.23E-1 | 5.42E+0 | 8.75E-1 | 3.57E+2            | 3.03E+4 | 2.58E+2 | 4.34E+3 | 7.00E+2  |
> > > | gpt2-m-DS    | 4.25E-3        | 9.78E-1 | 7.50E-3 | 1.35E-2 | 4.50E-3 | 1.15E-2            | 2.64E+0 | 2.03E-2 | 3.65E-2 | 1.22E-2  |
> > > | OPT1.3B      | 6.28E-1        | 6.28E+1 | 6.86E-1 | 1.07E+1 | 1.23E+0 | 5.02E+2            | 5.03E+4 | 5.49E+2 | 8.56E+3 | 9.82E+2  |
> > > | OPT1.3B-DS   | 1.02E-2        | 2.35E+0 | 1.80E-2 | 3.24E-2 | 1.08E-2 | 2.75E-2            | 6.33E+0 | 4.86E-2 | 8.75E-2 | 2.92E-2  |
> > > | OPT2.7B      | 1.94E+0        | 2.05E+2 | 2.11E+0 | 3.24E+1 | 4.01E+0 | 1.55E+3            | 1.64E+5 | 1.69E+3 | 2.60E+4 | 3.21E+3  |
> > > | OPT2.7B-DS   | 2.21E-2        | 5.08E+0 | 3.90E-2 | 7.02E-2 | 2.34E-2 | 5.97E-2            | 1.37E+1 | 1.05E-1 | 1.90E-1 | 6.32E-2  |
> > > | Llama2-7B    | 6.35E+0        | 6.28E+2 | 6.98E+0 | 1.09E+2 | 1.35E+1 | 5.08E+3            | 5.02E+5 | 5.58E+3 | 8.76E+4 | 1.08E+4  |
> > > | Llama2-7B-DS | 0.0561         | 12.903  | 0.099   | 0.1782  | 0.0594  | 1.51E-1            | 3.48E+1 | 2.67E-1 | 4.81E-1 | 1.60E-1  |

---

> > ### Comment · Reviewer_v14i · 2024-11-21
> >
> > Thank you for the clarification. The rebuttal has addressed my questions regarding stability, precision, and scalability. This is an intriguing and novel topic, and I am pleased to see this paper enabling further research in this direction.

---

> > > ### Author Response · Authors · 2024-11-21
> > >
> > > Thank you very much for your encouragement and positive feedback. We deeply appreciate your affirmation of our work. If there are any additional concerns or questions you’d like us to address to further enhance your confidence in this submission, please don’t hesitate to let us know.

---

### Official Review · Reviewer_kykQ · 2024-11-04

**Soundness:** 3
**Presentation:** 3
**Contribution:** 3
**Rating:** 6
**Confidence:** 3

**Summary:**

This work proposes a method for mapping LLM inference and training to Ising machines (which the authors refer to as "dynamical system machines"). They build upon existing work that introduces a room temperature Ising machine built with CMOS technology and work that maps graph learning problems to these substrates. The resulting DS-LLM is orders of magnitude more energy efficient than conventional hardware.

**Strengths:**

The problem tackled in this paper—the energy consumption of LLMs—is both timely and well-motivated. The approach taken by the authors is unique and the results appear quite significant. The methodology is clearly explained despite covering a broad range of background fields.

**Weaknesses:**

- Evaluation of inference improvements focus on a comparison against GPUs instead of low-precision, quantized implementations for edge devices. For example, comparing against low-precision CPU inference implementations (https://arxiv.org/pdf/2311.00502) or novel edge devices (https://arxiv.org/pdf/2409.15654, https://arxiv.org/pdf/2305.18691)
- There is no discussion of the possible limitations which is extremely relevant for ICLR readers as they may not be familiar with the hardware used in this paper. In particular, focus on scalability issues (like described in Question 1) or model flexibility issues (are there operations that would be difficult to map to the proposed substrate, and are all existing transformer variants mappable?).
- Small suggestion: the introduction would be clearer if the authors removed hyperbolic/unnecessary language surrounding the capabilities of natural annealing
    - e.g. Line 067: "Recently, Dynamical-System-based (DS) machines have emerged as a promising solution of such, which leverages nature itself as a computer!"
    - Paragraph on Lines 077-090 can be shortened as it mostly restates the previous paragraph with longer phrasing. Instead, simply state that no existing framework maps LLMs to DS machines, the prior work is specific to graph learning, and the difference in computation and scale between LLMs and graph learning poses a challenge for finding such a mapping.

**Questions:**

1. The prior work on NP-GL utilized relatively small graphs, resulting in circuits with 100s of nodes. This is not the case for a transformer model where the number of nodes is much larger. Is it possible to manufacture Ising machines at this scale? If not, would we need to rely on a multi-chip solution?
2. The description of back-propagation implemented on the Ising machine is not clear. What is computed by the auxiliary circuit and what is computed by the Ising machine? Is there a separate "reverse" network for computing gradients? If so, how are the reverse and forward networks kept aligned (typically referred to as the "weight transport problem" in brain-inspired learning settings)?
3. Does the time and energy evaluation refer to the total training / inference time over the full dataset or a single sample?
4. Does your proposed hardware solution perform training with batches? If not, wouldn't the total training time be prohibitive?

---

> ### Author Response · Authors · 2024-11-21
> **Part 1 of Response to Reviewer kykQ**
>
> ## Overview:
>
> Thank you very much for your insightful and meaningful feedback, we have uploaded the revised paper to address your concerns and you can find the contents according to the lable for Reviewer B like "B-Q1". We would like to list our detail response to your questions as below:
>
> ## 1. Comparison with implementations for edge devices (weakness 1):
>
> We sincerely appreciate you for your valuable suggestion. We have included a new comparison in the revised version to evaluate our work against low-precision CPUs [1] and emerging edge devices [2] on the Llama2-7B model. This model was specifically chosen as both referenced studies evaluated it in their work, allowing for a direct and meaningful comparison. For power estimation of the CPU work [1], the authors did not provide the exact power consumption of the “4th Generation Intel Xeon Scalable Processors” used. Hence, we conservatively assumed a power consumption of 100W, which is lower than any product in the stated CPU series. The results show that DS-LLM achieves orders of magnitudes higher token generation rate and energy efficiency than the references. This is because both the low precision CPU works and edge devices are still digital computing diagram which follows a step-by-step instructions-based computing, while DS machines don’t need micro instructions and perform continuous natural annealing by analog currents which is extremely fast and energy efficient.
>
> | Solutions | token/s | token/KWh |
> |:---------------------:|:-----------:|:--------------:|
> | Low precision CPU [1] | 45.4 | 1.63E+06 |
> | Cambricon-LLM [2] | 3.55 | 3.60E+06 |
> | DS-LLM （this work) | 3.03E+04 | 4.04E+10  |
>
>
> ### Reference:
> [1] Efficient LLM Inference on CPUs (https://arxiv.org/pdf/2311.00502)
>
> [2] Cambricon-LLM: A Chiplet-Based Hybrid Architecture for On-Device Inference of 70B LLM (https://arxiv.org/pdf/2409.15654)
>
> ## 2. Hardware limitations (weakness 2 and question 1):
>
> We sincerely appreciate your insightful comments, which help us a lot to improve this work. We have added a discussion section in the Appendix to address your concerns on hardware limitations including scalability and model flexibility.
>
> ### Scalability Issues:
>
> While prior work like NP-GL was designed for small graphs with fewer than 1,000 nodes, the capacity of DS machines can be significantly expanded to handle much larger scales. First, DS machines have demonstrated linear complexity with respect to the number of nodes [3], making them inherently scalable with increased chip area. For context, NP-GL occupies only about 5 mm², whereas modern GPUs like the H100 have a die size of 814 mm², and wafer-scale chips—such as those with up to 46,255 mm²—are emerging [4]. Based on linear complexity, a single-chip solution could theoretically support millions of nodes within a single DS machine.
>
> Second, for even larger models or faster training, multi-chip approaches offer a viable path forward. Existing research has explored multi-chip solutions for DS machines [5], where individual chips perform annealing with periodic synchronization. We are also investigating promising techniques such as deploying models across multiple DS machine chips to achieve pipeline parallelism.
>
> Overall, the scalability of DS machines is theoretically well-founded, offering both single-chip and multi-chip solutions to meet the demands of increasingly large and complex models.
>
> ### Model Flexibility Issues:
>
> This work focuses on classic operations in Transformer-based LLMs, acknowledging that modern LLMs may include different operations such as varied activation functions or embedding methods. Despite the diversity of LLM architectures, these operations can generally be categorized as either polynomial or non-polynomial. Polynomial operations, which can be broken down into basic addition and multiplication, are directly mappable to DS machines. Non-polynomial operations, such as exponential function, require transformation into polynomial approximations (e.g., via Taylor expansion) or the addition of auxiliary circuits, which may slightly increase latency depending on their complexity. Fortunately, most high-computational-demand operations, particularly in attention layers and FFNs, are polynomial or even linear. Thus, DS machines offer high flexibility and adaptability for various models.
>
> ### Reference:
>
> [3] DS-GL: Advancing Graph Learning via Harnessing Nature’s Power within Scalable Dynamical Systems
>
> [4] Wafer-Scale Computing: Advancements, Challenges, and Future Perspectives [Feature]
>
> [5] Increasing ising machine capacity with multi-chip architectures
>
> ## 3.Introduction revision (weakness 3):
>
> Thank you for your helpful suggestions. We have revised the introduction to make it clearer.
>
> -------------------
> *Due to character limitation, other responses wil be listed in next comments.

---

> > ### Author Response · Authors · 2024-11-21
> > **Part 2 of Response to Reviewer kykQ**
> >
> > ## 4. Clarification of “reverse” network (question 2):
> >
> > Thank you very much for this insightful question. We have updated figure 5 and related contents in the revised paper to clarify the details about the back-propagation part of DS machine. We don’t use a “reverse” network for the computing of gradients but load the activations and calculate the gradients in standard way. For most operations, this calculation is also a matrix multiplication operation and thus can be mapped into the DS machines. Only non-polynomial operations will be processed by auxiliary circuits.
> >
> > ## 5. Time and energy evaluation (question 3):
> >
> > We appreciate a lot for your valuable comments, and we have added more detail about the setup. The time and energy evaluation refer to the total training / inference time over the full train_set / eval_set.
> >
> > ## 6. Batch Training (question 4):
> >
> > Thank you for this meaningful question. The proposed hardware solution supports batch training. Additionally, there is significant exploration potential at the system integration level for deploying distributed training across multiple DS machines, like multi-GPU setups. This represents a promising direction for future research.

---

> > > ### Author Response · Authors · 2024-11-25
> > >
> > > We deeply appreciate the time and effort you have dedicated to reviewing our work and for providing such constructive feedback. For your reference, we have recently uploaded a new revision to further enhance the paper, with the latest modifications marked in blue and previous revisions marked in green.
> > >
> > > As the discussion period draws to a close, we would greatly appreciate it if there were any feedback or suggestions regarding our revision.
> > >
> > > Ensuring that we have adequately addressed your concerns remains our top priority, and any further insights or thoughts from you would be immensely valuable in refining and improving our work. Again, thank you for your thoughtful engagement and continued support throughout this process.

---

> > > > ### Comment · Reviewer_kykQ · 2024-11-25
> > > >
> > > > Thank you to the authors for addressing my concerns. The overall quality of the writing and evaluation has been improved substantially. I have noted the following changes:
> > > > - the introduction is much clearer and more concise (and the added space is used to refer to prior work—which is good)
> > > > - figures and tables added to address the scaling properties of the proposed hardware
> > > > - evaluation against realistic baselines for energy-efficient inference
> > > >
> > > > Accordingly, I am raising my score, erring on the side of acceptance given the impressive results. My reason for not giving an even higher score is similar to Review HAWr. While this work shows a lot of promise, its primary contribution is translating LLM ops to DS hardware primitives. This may be the first work to do this, and it might be important for ongoing research applying DS machines to LLMs. However, with respect to the ICLR audience, I do not feel this work adds novelty substantially beyond the prior work. Instead, the target audience might be at a hardware conference (e.g., MICRO, ISCA).
> > > >
> > > > I have some final minor suggestions that the authors should consider for the final revision.
> > > > - Fig. 7 has an "estimated" section of the curves, but the text only mentions that estimation is done "by roughly projection". I suggest being more specific about exactly how the points are estimated from the existing data. It could be as simple as "linearly extrapolated from X" (if that's what was done).

---

> > > > > ### Author Response · Authors · 2024-11-25
> > > > >
> > > > > We greatly appreciate your recognition of our efforts and are delighted to hear that our revisions have satisfactorily addressed your concerns. Your thoughtful and constructive feedback has been invaluable in shaping our work, and we are truly grateful for your acknowledgment of its significance. Thank you again for your final suggestion and we will update our revision as soon as possible.

---

> > > > > ### Author Response · Authors · 2024-11-26
> > > > >
> > > > > We have made further revisions based on your suggestion. Again, we sincerely thank you for your support and help in strengthening our paper.

---

> ### Author Response · Authors · 2024-11-26
>
> The discussions over the past few days have been incredibly insightful and have greatly contributed to improving our paper. Much appreciated for the reviewer's suggestions. With the extension of the discussion period, we see this as a rare and invaluable opportunity to further engage with the reviewer and learn from their expertise and wisdom.
>
> Therefore, we would like to take this opportunity to clarify the new contributions of this work compared to prior art (e.g., NP-GL), beyond the mapping of LLM kernels onto DS machines. Specifically, DS-LLM (this work) represents the first effort to enable on-device training on a dynamical systems (DS) machine, thereby extending the computational capabilities of DS machines from inference to training. In contrast, prior works restricted DS machines to inference tasks, requiring training to be conducted on GPUs.
>
> Once again, we deeply appreciate the reviewer’s recognition of the potential of our work in advancing LLM development and their constructive feedback throughout this process. We are always eager to receive any additional suggestions.

---

### Official Review · Reviewer_xU44 · 2024-11-05

**Soundness:** 2
**Presentation:** 3
**Contribution:** 3
**Rating:** 6
**Confidence:** 3

**Summary:**

The paper introduces DS-LLM, a novel framework that employs Dynamical System (DS)-based machines to optimize the training and inference processes for Large Language Models (LLMs). By incorporating an electrodynamics-based model and a Natural Annealing process, DS-LLM dramatically reduces the computational costs associated with these models. The approach is mathematically and empirically validated, demonstrating significant speedups and energy reductions without sacrificing accuracy compared to conventional LLMs.

**Strengths:**

1. The paper pioneers the use of DS machines in the context of LLMs, leveraging their inherent Natural Annealing processes to optimize both training and inference, which is both novel and potentially transformative for the field.
2. The reported results show a 1,090× speedup and a 102,559× reduction in energy during training, along with a 127× speedup and a 37,545× reduction during inference. These improvements are substantial and address critical cost and efficiency barriers in LLM deployment.
3. The evaluation is thorough, with tests across multiple models and datasets demonstrating that DS-LLM achieves comparable accuracy to traditional LLMs. This thorough testing helps validate the practicality of the proposed approach.

**Weaknesses:**

1. While the paper demonstrates impressive theoretical and simulated results, the practical implementation of such systems might be challenging due to specialized hardware requirements. To better understand the feasibility, could you discuss specific hardware requirements, potential adoption challenges, and necessary adaptations to existing ML infrastructures for implementing DS-LLM?
2. The paper discusses the theoretical and simulated benefits of DS machines but lacks concrete examples of these machines being implemented in real-world scenarios.

**Questions:**

1. Can you provide specific examples of Dynamical System-based machines being used in real-world applications?
2. What are the main challenges in integrating DS machines with the existing computational infrastructures used for LLMs?
3. Could you provide more specifics about the Finite Element Analysis (FEA) software emulator used for the DS machine simulations?
4. How closely do the results from the FEA simulations align with what you would expect from real hardware implementations? Are there any known discrepancies or performance variations between the simulated outcomes and actual hardware tests?

---

> ### Author Response · Authors · 2024-11-21
> **Response to Reviewer xU44**
>
> ## Overview
> We sincerely thank you for the valuable suggestions, which have significantly helped us improve the quality of this paper. We have uploaded a revised version addressing the points you raised and you can find the contents according to the label for reviwer A like "A-Q1". The specific response to your questions are as below:
>
> ## 1. Examples of real-world applications (weakness 2 & question 1):
>
> Thank you for your suggestion and we have incorporated concrete examples of DS machines being applied to real-world problems in the introduction of the revised manuscript. The applications include traffic predictions in [1], air quality, taxi demand, and pandemic progression in [2], and optimization problems like MAX-CUT in [3].
>
> ### References:
> [1] Ising-Traffic: Using Ising Machine Learning to Predict Traffic Congestion under Uncertainty.
>
> [2] Extending Power of Nature from Binary to Real-Valued Graph Learning in Real World.
>
> [3] Experimental investigation of performance differences between coherent Ising machines and a quantum annealer.
>
> ## 2. Feasibility of DS machines (weakness 1 & question 2):
>
> Thank you very much for this insightful feedback and we have added a discussion section in Appendix to analysis the feasibility of DS machines in detail. While this work is early-stage research exploring the potential of DS machines to meet the growing computational demands of LLMs, we would like to discuss the following key points address their feasibility and future integration:
>
> First, we want to highlight that there are no fundamental challenges to integrate DS machines into existing computing systems. DS machines, though architecturally distinct from digital processors, are built using CMOS-compatible technology. This compatibility ensures they can be integrated seamlessly into existing systems as co-processors (similar to TPUs or NPUs) via interfaces like PCIe. Theoretically, no fundamental hardware adaptations are required.
>
> From a system perspective, though this work is still in early stage, we agree that integration to the existing computing infrastructure is very promising as future exploration. There are a lot of exploration space on developing software toolchain like compilers, optimizing memory management, and pipelining tasks between DS machines and other processors. With all these works and software tools help, we believe DS machines are inherently feasible for integration into digital systems and work as a new type of co-processor like GPUs / TPUs / NPUs. Future work can explore hybrid use cases that combine CPUs, GPUs, and DS machines based on their unique strengths to achieve optimal performance.
>
> Overall, while DS machines are not yet mature, their fundamental feasibility paves the way for exciting opportunities in integration with existing computing infrastructures.
>
> ## 3. Detail of the FEA emulator (question 3):
>
> Thank you for your meaningful suggestion. The FEA software emulator we used is adapted from the one used in BRIM, which has been validated to provide results comparable to actual chip prototypes. EFA is a SOTA simulation approach for circuit-level simulation that incorporates the physical parameters such as capacitance, resistance, and temperature, simulating voltage and current evolution in tiny time steps (100 ps). In essence, it functions as a GPU-accelerated poor-man’s circuit-level simulation tool for Natural Annealing processes.
>
> ## 4. Error of FEA simulation results (question 4):
>
> Thank you for raising this insightful point. While our FEA simulation assumes ideal conditions, it is worth noting that actual chips, including conventional CPUs/GPUs, often encounter non-ideal factors during manufacturing. We have not yet manufactured a prototype for this specific work, but the evaluation results on a prototype for previous work shows a less than 1% error between the simulation and actual chip. The previous prototype was designed for graph problems but fundamentally has the same basic components as this work. Therefore, we believe the judgement can be extended to this work that the circuit-level EFA simulation is a stable approach for DS machines.

---

> > ### Author Response · Authors · 2024-11-25
> >
> > We sincerely appreciate the time and effort you have invested in reviewing our work and providing such valuable feedback. We have recently uploaded a revised version of the paper, with the latest modifications highlighted in blue and prior changes marked in green.
> >
> > Please feel free to let me know if there were any feedback or suggestions regarding our revision. Ensuring that we have adequately addressed your concerns remains our top priority, and any further insights or thoughts from you would be immensely valuable in refining and improving our work. Again, thank you for your thoughtful engagement and continued support throughout this process.

---

### Author Response · Authors · 2024-11-21
**Thank you to all reviewers**

### We sincerely thank all the reviewers for their insightful and constructive feedback, which has greatly helped us improve the quality of this paper. We have uploaded a revised manuscript and labeled it corresponding to reviewer questions. Due to space constraints, most detailed responses to the comments are added in the Appendix.

### Here is a summary of the major revisions:
1.	Comprehensive Discussion on DS Hardware: We have included an extensive discussion in the Appendix covering aspects such as stability, feasibility, robustness, scalability, non-idealities, and potential challenges, along with future exploration directions.
2.	Extended Model Evaluation: We have added evaluations on larger models, up to 7B, to better demonstrate the scalability and effectiveness of our solution.
3.	Clarifications and Experimental Details: More details have been provided to clarify key concepts and experimental setups for better transparency.
4.	Fairer and Clearer Comparisons: Additional experimental results have been added to ensure fair and clear comparisons.

### We also want to emphasize that this work, as a cutting-edge exploration, introduces a promising solution to address the growing computational demands of LLMs. While DS machines are not yet mature for commercial applications, they hold immense potential to become a critical component in next generation computing paradigms, fundamentally tackling the challenges of LLM computing.

### Once again, we deeply appreciate the reviewers for their time and invaluable efforts. Please feel free to reach out if you have any further questions regarding the revisions. Thank you!

---

### Author Response · Authors · 2024-12-03
**Part2 of Summary of Rebuttal and Discussion**

We would like to emphasize the contributions of DS-LLM as recognized by reviewers:

1. *Introducing a new computing paradigm to LLM*: DS-LLM is the first to losslessly equate neural network (NN) kernels to the natural annealing processes of DS machines, transforming expensive and explicitly programmed NN computation into significantly more efficient nature-powered processes. By achieving this lossless equivalence, DS-LLM extends the computing power of DS machine to complex NN models including large language models (LLMs).
2. *On-device training support*: DS-LLM is the first to enable on-device training on a DS machine, demonstrating that DS machines have the ability to perform not only inference but also efficient training, bringing the computing power of DS machine to LLM training. In contrast, prior works restricted DS machines to inference tasks, requiring all parameter training still to be conducted on GPUs.
3. *Significantly reducing LLM computing cost*: DS-LLM achieves orders of magnitude speedup and energy reduction over traditional GPU baselines, showcasing the impressive potential to significantly reduce the cost of LLM computing on both inference and training.

We extend our sincere gratitude to the reviewers for their invaluable feedback, which has greatly improved the quality of this work. We hope DS-LLM provides meaningful insights and serves as a foundation for further exploration in this promising area.

Sincerely,

Authors

---

### Author Response · Authors · 2024-12-03
**Part1 of Summary of Rebuttal and Discussion**

### Dear Area Chair and Reviewers,

We deeply appreciate the constructive feedback and insightful suggestions provided throughout the review process. Your thoughtful comments have been instrumental in strengthening our work. We are particularly encouraged by the positive consensus across all reviewers during the discussion phase:

1. Reviewer xU44 retained their positive score. Their comments are very insightful and constructive, and we have addressed each point thoroughly. Our responses have been incorporated into the revised manuscript.
2. Reviewer kykQ raised their rating, recognizing the enhancements made to our introduction, the comprehensive analysis of hardware scalability, and the comparison with realistic baselines. They also highlighted the potential importance of this work for ongoing research in applying DS machines to LLMs.
3. Reviewer v14i maintained their positive score, appreciating our efforts to address concerns about stability, precision, and scalability. They commended the work as intriguing and novel, expressing enthusiasm about its potential to inspire further research.
4. Reviewer HAWr significantly increased their score, acknowledging the improved discussions on limitations, enhanced baselines and metrics, and more detailed analyses of scalability and nonidealities. They noted that this work is both interesting and likely the first to tailor DS machines specifically for LLMs.

As the discussion concludes, we summarize below the key revisions made in response to your feedback:
### 1. Enhanced Related Works
To address Reviewer xU44’s request for more real-world context, we added concrete examples in the introduction illustrating DS machines’ applications to practical problems.

In response to Reviewer HAWr, we expanded the introduction to differentiate our approach from prior work. In Appendix Sec. A.1, we elaborated on the unique advantages of DS-LLM compared to other emerging computing technologies, providing a nuanced perspective on its contributions to the field.

### 2. Discussion on Hardware limitations
Following Reviewer xU44’s recommendation, we analysis and discuss the feasibility of DS machines in a new added discussion section in Appendix Sec. A.1, highlighting the fundamental feasibility of DS machines on integration with existing computing infrastructures.

In response to Reviewer kykQ’s request, we included a new section in Appendix Sec. A.1 discussing the scalability and model flexibility issues. We provided supportive reference for the scalability potential of DS machines and highlighted the inherent model flexibility by theoretically analysis.

Reviewer v14i raised concerns on limitations of DS machines specifically on stability, precision, and scalability issues. We provided a new experiment demonstrating DS machines' high stability and discussed precision and scalability challenges and solutions in detail within Appendix Sec. A.1.

Following Reviewer HAWr’s request, we revised the abstract, introduction, and evaluation section to reflect the potential challenges on DS machines and solutions, providing a reliable and practical analysis of the approach. We also added a new discussion section as Sec. 5 to highlight the challenges and solutions on physical practicality and system integration of DS machines, and we addressed other discussions on stability, non-idealities, and trade-offs on Taylor expansion in Appendix Sec A.1. Additionally, we provided a better visualization with theoretical projections on scaling curve on training/inference performance in Fig. 7, with detailed cost of scaling in Table. 3, showing a comprehensive analysis of the scalability of our approach.

### 3. Improvements on model evaluation
Reviewer kykQ suggested benchmarking against low-precision, quantized implementations for edge devices. To this end, we included a comparison with low-precision CPUs and emerging edge devices, showing that DS machines outperform these solutions.

As requested by Reviewer v14i, we included evaluations on two larger-sized models including OPT-2.7B and Llama2-7B to better assess our solution’s performance. The results in Table. 1 and Table. 2 demonstrate that DS-LLM continues to offer comparable accuracy and significant speedup & energy reduction.

Following the suggestion from Reviewer HAWr, we have incorporated vLLM to enhance GPU inference performance and updated Table. 2 with metrics such as token/s and token/KWh to provide a fairer and more comprehensive comparison.

### 4. Clarification on experimental details
Addressing Reviewer xU44’s request, we expanded the experimental setup section to include more details on the FEA emulator and its stability.

Addressing concerns from Reviewer uXT1, we added detailed explanations of time and energy evaluations and clarified batch training capabilities.

Reviewer KNeJ sought clarification on experimental setup and metrics. We have incorporated these details and updated the metrics based on their suggestions.

---

### Meta-Review · Area_Chair_kzAP · 2024-12-22

**Metareview:**

This work introduces DS-LLM, a framework that employs Dynamical System (DS)-based machines to optimize Large Language Models (LLMs), significantly reducing computational costs through an electrodynamics model and Natural Annealing process. The approach achieves substantial speedups and energy reductions without compromising accuracy. The paper demonstrates the potential of DS machines to transform LLM training and inference, offering a promising path toward more efficient and sustainable AI model deployment.
All the reviewers give positive comments on this work, thus I recommend the acceptance of this work.

**Additional Comments On Reviewer Discussion:**

The concerns of the reviewers have been well-addressed.

---

### Decision · Program_Chairs · 2025-01-22

Accept (Poster)